# Human decision making balances reward maximization and policy compression

**Lucy Lai**[1,2]*, **Samuel J. Gershman**[3]

**1** Program in Neuroscience, Harvard University, Cambridge, Massachusetts, United States of America,
**2** Theoretical Sciences Visiting Program, Okinawa Institute of Science and Technology Graduate University,
Onna, Okinawa, Japan, **3** Department of Psychology and Center for Brain Science, Harvard University,
Cambridge, Massachusetts, United States of America

* lucylai@g.harvard.edu

## Abstract

Policy compression is a computational framework that describes how capacity-limited agents trade reward for simpler action policies to reduce cognitive cost. In this study, we present behavioral evidence that humans prefer simpler policies, as predicted by a capacity-limited reinforcement learning model. Across a set of tasks, we find that people exploit structure in the relationships between states, actions, and rewards to "compress" their policies. In particular, compressed policies are systematically biased towards actions with high marginal probability, thereby discarding some state information. This bias is greater when there is redundancy in the reward-maximizing action policy across states, and increases with memory load. These results could not be explained qualitatively or quantitatively by models that did not make use of policy compression under a capacity limit. We also confirmed the prediction that time pressure should further reduce policy complexity and increase action bias, based on the hypothesis that actions are selected via time-dependent decoding of a compressed code. These findings contribute to a deeper understanding of how humans adapt their decision-making strategies under cognitive resource constraints.

## Author summary

Decision making taxes cognitive resources. For example, when shopping for groceries on a budget, we must evaluate which brand offers the best value for the price. But time constraints or mental fatigue can often steer us towards familiar choices, such as sticking to the same brand. To understand how cognitive resource limitations affect human decision making, we conducted a study in which we manipulated the number of optimal choices and the time limit within which choices were made. Across three tasks, we found that people utilize task structure to compress the amount of information factored into their decision making. Information compression biases people towards their past choices. This bias persists even when multiple optimal choices are available, and intensifies under cognitive load and time pressure. A computational model of decision making under cognitive

---

**Data Availability Statement:** All data and code used for experiments and analysis are available at https://github.com/lucylai96/simplepolicies.

**Funding:** This research was supported by a Harvard Brain Science Initiative Bipolar Disorder

Seed Grant to SJG, (https://brain.harvard.edu/psychiatric-disorders/); the National Science Foundation Graduate Research Fellowship, DGE-1745303 to LL, (https://www.nsfgrfp.org/); and the 28Twelve Foundation Harvey Fellowship to LL, (https://www.28twelvefoundation.org/). The funders had no role in study design, data collection and analysis, decision to publish, or preparation of the manuscript.

**Competing interests:** The authors have declared that no competing interests exist.

constraints accurately describes the experimental data. Our findings may have the potential to inform the design of choice environments that better align with human decision biases.

## Introduction

Everyday decision making requires individuals to learn about the relationship between states and actions in their environment. For example, when deciding what to eat for lunch (the action), a decision-maker might consider the nutritional content as well as the price of a meal (information about the state). They may aim to maximize the overall value of their choices, such as choosing the cheapest, calorie-dense option. But because decision making consumes cognitive resources, the quality of our decisions is often bounded by the availability of those resources. The "resource-rational" approach suggests that people are making the best choices they can subject to constraints on their cognitive resources [1, 2]. These resources can be formalized in terms of the physical systems that implement decision making. For example, all finite physical storage systems, such as the brain, have memory constraints that limit their capacity to store and transmit information. This capacity limit has important implications for, and effects on, decision making.

In the reinforcement learning (RL) framework, the decision-maker learns a *policy* that maps states of the world to actions [3]. When cast in the language of information theory, which provides a formalism for understanding how information is stored and transmitted, policies can be viewed as communication channels that transmit information about state (the input) to guide action selection (the output; Fig 1A) [4]. The capacity of a channel is defined as the maximal mutual information between its inputs and outputs (also known as the *rate*), and thus we refer to the mutual information between states and actions as the agent's *policy complexity*. The capacity of the action selection channel is thus the maximal policy complexity that the agent can achieve (Fig 1D).

Policy complexity measures the amount of information about state used to select actions, or how much an agent "pays attention" to the state. Paying more attention to the state allows the agent to make reward-maximizing decisions, but the channel capacity limits the agent's ability to encode states with high fidelity, thus impacting decision making. This implies a trade-off between policy complexity and task performance. Limited capacity forces the agent to "compress" their policies (i.e., reduce their state-dependence), thereby reducing performance [5–11]. We refer to this as the "policy compression" framework.

Recent work has used policy compression to explain a wide range of decision making phenomena, such as perseveration [12], cognitive deficits in schizophrenia [10], mouse navigation [13], and undermatching [14], among other examples. However, all of these studies have relied on *post hoc* analyses of data from previously published research. In the current study, we use theory to guide experimental design in order to directly test the unique, key predictions of the policy compression framework. This approach also allows us to explore previously untested hypotheses about learning under cognitive constraints. For example, while some previous work focused on applying policy compression to understand set size effects (a decrease in performance with the number of distinct stimuli to remember; [10, 12]), we ask whether policy compression can systematically vary even when set size is held fixed. Furthermore, [12] used policy compression to provide a normative explanation for perseveration, the tendency to produce the same action policy across states, irrespective of the reward outcome. By intentionally designing the distribution of rewarded actions, we can

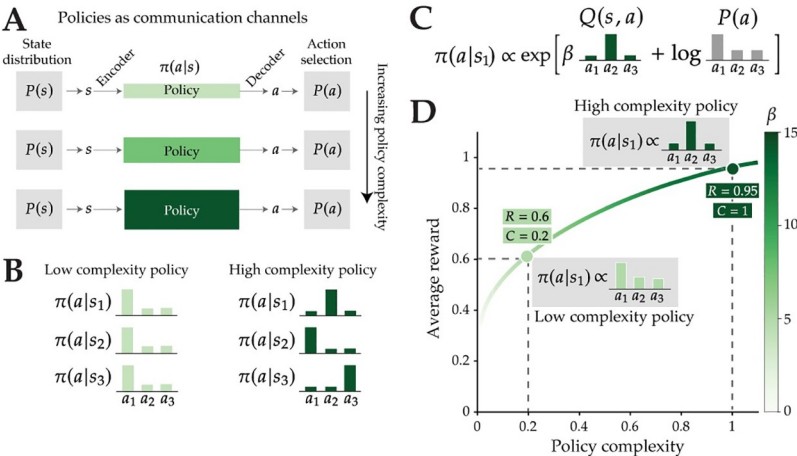

**Fig 1. Capacity-limited action selection. (A)** Policies can be viewed as capacity-limited channels that communicate information about states to guide action selection. A state distribution $P(s)$ generates states $s$ that are encoded into memory. The policy complexity (depicted as the size of the channel) is the mutual information (in bits) between states and actions, or the amount of information from the state that is used for action selection. **(B)** Policy complexity is low when the distribution over actions is the same in each state (left), and it is high when the action distribution is different for each state (right). **(C)** The optimal policy combines state-action values $Q(s, a)$ with a marginal action probability term $P(a)$ that biases the policy towards actions that are chosen frequently across all states. The trade-off term, $\beta$, determines the relative contribution of $Q(s, a)$ and $P(a)$, thereby controlling how state-dependent action selection is. Example distributions are shown to depict action selection in one state. **(D)** A limit on the channel capacity (or a set aspiration level) results in a trade-off between reward and complexity. The $\beta$ parameter increases monotonically with policy complexity. Using the depicted example in (C), two policies with different complexities are shown, along with the agent's theoretical capacity limit, $C$, and aspiration level, $R$. The light green point on the curve illustrates a low complexity policy (low $\beta$), resulting in a distribution of actions that closely resembles the marginal distribution $P(a)$. The dark green point on the curve illustrates a high complexity policy (high $\beta$), resulting in a distribution of actions that more closely resembles the state-action values $Q(s, a)$.

directly test the prediction that individuals with low policy complexity will perseverate optimally—that is, their choices will be biased towards actions they have chosen most frequently in the past.

In the current study, we hypothesize that structure in the probabilistic relationships between states, actions, and rewards will shape how agents compress their policies. To test this hypothesis, we design tasks that manipulate the distribution of states and actions in ways that encourage policy compression. Across three tasks, we show that people adjust their policy complexity in response to the characteristics of their environment. We find that people consistently prefer simpler policies, exploiting structure in the distribution of states and the redundancy of actions across states to compress their policies. As a result, choice behavior is systematically biased towards actions with higher marginal action probabilities, as predicted by policy compression models. Moreover, we show that individuals reduce their policy complexity under time pressure, providing evidence for the hypothesis that actions are selected through time-sensitive decoding of a compressed code [11]. Our results cannot be explained by models that do not compress policies under a capacity constraint, including those that consider working memory contributions to reinforcement learning [15]. Taken together, these results provide strong support for policy compression models, illuminating how individuals leverage the structure of the environment to simplify their policies.

## Results

### Modeling overview

We briefly summarize the main components of the policy compression model that are directly relevant to the subsequent behavioral experiments. A more comprehensive presentation of the theory can be found in the Methods section.

In the reinforcement learning (RL) framework, the agent's goal is to learn an optimal policy $\pi^*$ that maximizes expected reward:

$$\pi^* = \underset{\pi}{\mathrm{argmax}} \ V^\pi, \tag{1}$$

where $V^\pi$ is the expected reward earned by following policy $\pi$:

$$V^\pi = \sum_s P(s) \sum_a \pi(a|s) Q(s, a). \tag{2}$$

Here, $P(s)$ is the probability of state $s$, and $Q(s, a)$ is the expected reward in state $s$ after taking action $a$. A capacity-limited agent faces the additional constraint that its policy complexity (the mutual information between states and actions) cannot exceed its capacity $C$. The optimal policy for such an agent that seeks to maximize expected reward subject to their capacity limit is [5, 6, 8, 11]:

$$\pi^*(a|s) \propto \exp[\beta Q(s, a) + \log P^*(a)], \tag{3}$$

which is a softmax function that combines state-action values $Q(s, a)$ with a marginal action probability term $P^*(a) = \sum_s P(s)\pi^*(a|s)$ that biases the policy towards actions that are chosen frequently across all states. The optimal policy results in a trade-off between average reward and the agent's policy complexity (Fig 1D). This trade-off is mediated by the inverse temperature parameter term, $\beta$, which indexes how state-dependent a policy is: When $\beta$ is close to 0, the policy will be state-independent, driven by actions that are overall chosen more frequently (the $P^*(a)$ term). As $\beta$ increases, the policy will select actions that yield the most reward, conditional on the current state (the $Q(s, a)$ term; see Fig 1D for a visual example). In summary, policy complexity is higher when the policy depends strongly on the state: it is maximized when each state maps to a unique action, and it is minimized when the distribution over actions is the same in each state (Fig 1B). Finally, we note that another way to view policy compression is to minimize policy complexity subject to a fixed aspiration level $R$ (an agent's desired reward rate; see [8]). The two optimization problems (maximizing reward and minimizing complexity) can lead to the same optimal policy if the aspiration level $R$ is chosen to be the highest expected reward achievable under capacity $C$ (Fig 1D).

Because of the influence of the marginal action distribution, a capacity-limited agent's policy and subsequent behavior may be significantly influenced by the task structure, compared to a standard reinforcement learning (RL) agent whose policy is:

$$\pi(a|s) \propto \exp[\beta Q(s, a)]. \tag{4}$$

To illustrate this, imagine a task where there are three states and three possible actions to choose from. Two of the states share the same rewarded action, while the last state has its own uniquely-rewarded action (Fig 2A). In the policy compression model, the agent's choice behavior in each state is biased by the marginal distribution of actions, which can be seen in the predicted choice probabilities of suboptimal actions with non-zero marginal probability. This is especially true for agents with low capacity constraints (Fig 2B, light green). Notably, a Standard RL agent matched for the same value of $\beta$ is not biased by the marginal action

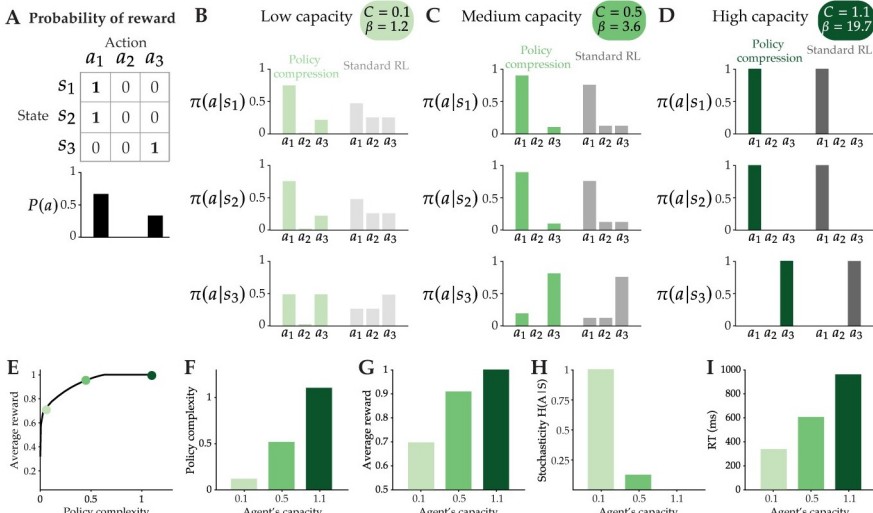

**Fig 2. Task structure influences choice behavior. (A)** In this example task, two states share one, deterministically-rewarded optimal action, and one state has its own uniquely-rewarded action. The marginal action distribution $P(a)$ depicted is derived from the optimal unbounded policy, which assumes that $\beta \to \infty$ (i.e., assuming that subjects perfectly learn the reward function) and is biased towards $a_1$. **(B)** For an agent with a low capacity limit and therefore low $\beta$, the policy is similar across states and closely resembles $P(a)$. A Standard RL agent's policy under the same $\beta$ value is not influenced by the marginal distribution of actions. **(C)** Same as (B) for an agent with a medium capacity limit. While the state-dependent policies become more "peaked" on the optimal action, the policy compression model still predicts some influence of $P(a)$, particularly on the choice probabilites of suboptimal actions with non-zero marginal probability. Suboptimal actions are chosen equally under a Standard RL agent's policy with the same $\beta$ value. **(D)** Same as (B) for an agent with a high or unbounded capacity limit (in this case they are equivalent, since the agent's capacity limit is equal to the task complexity). Under a sufficiently high value of $\beta$, the policy compression and Standard RL agent make the same predictions. **(E)** The reward-complexity trade-off curve for the task shown in (A). Each colored point indicates the performance of an optimal agent with a low ($C = 0.1$), medium ($C = 0.5$), or high ($C = 1.1$) capacity limit (from left to right). **(F)** The optimal policy complexity of each agent depicted in (E). **(G)** The average reward obtained by the policies learned by each agent in (E). **(H)** The stochasticity of agents' choices, measured as the conditional entropy of action given states. **(I)** Example response times (RT) for each agent, generated from a policy compression model that assumes a linear relationship between policy complexity and RT.

distribution. While there is more stochasticity across actions (reflecting the low $\beta$), the Standard RL agent chooses suboptimal actions with equal probability. For the capacity-limited agent, the bias towards the marginal action distribution is gradually reduced by increasing capacity (Fig 2C). At high enough capacities (namely, capacity limits that are at or exceed the task complexity), the bias disappears, and the policy compression model and Standard RL model become equivalent (Fig 2D). An agent's capacity limit places an upper bound on its policy complexity (Fig 2F), which determines the maximum average reward that can be earned (Fig 2G), and affects other aspects of choice, such as stochasticity (Fig 2H) and response time (Fig 2I).

In our previous work [11], we argued that response time (RT) should be a linear function of policy complexity, which can be manipulated even when the number of states is held fixed [16]. Consistent with this prediction, we found that lower policy complexity significantly predicted shorter response times in a contextual multi-armed bandit task [11, 17]. In the current study, we directly test the relationship between complexity and RT by building trial-by-trial RT predictions into our model. To do this, we make two assumptions. First, we assumed that RT is monotonically related to the policy cost, which indexes the cost of taking a specific action $a$ in the current state $s$ by quantifying the deviation of the state-specific action policy $\pi(a|s)$

from the marginal action probability $P(a)$: $\log \frac{\pi_\theta(a|s)}{P(a)}$. This implies that the RT will be slower in trials where the agent selects an action with a higher state-specific probability relative to its marginal probability. Second, we assumed that RT is monotonically related to the entropy of the policy on a given trial:

$$H = -\sum_a \pi_\theta(a|s) \log \pi_\theta(a|s). \tag{5}$$

This assumption is based on the idea that greater "action uncertainty" (i.e., more dispersed policies) should produce slower RTs [18–20]. For example, if the action probabilities are all similar (i.e., if $\pi(a|s)$ is roughly uniform), we should expect high uncertainty and a slow RT. This assumption captures the overall decrease in RT due to learning, as policies tend to become more "peaked," or lower in entropy, with learning. Note that policy complexity alone cannot capture this pattern, because in some conditions policy complexity increases over the course of learning while RTs continue to decline. Using these two quantities, we can specify a regression model relating policy cost and entropy to response time (in milliseconds) [21]:

$$\log \text{RT} = \log \left[ t_0 + b_1 \left( \log \frac{\pi_\theta(a|s)}{P(a)} \right) + b_2 H \right] + \epsilon \tag{6}$$

where $t_0$ is non-decision time and $\epsilon \sim N(0, \sigma^2)$ is Gaussian random noise.

We considered several variants of our policy compression model, which vary in their number of free parameters: "Fixed" models assume that the $\beta$ parameter remains constant throughout learning, while "Adaptive" models assume that $\beta$ is updated according to the dynamics of learning. Within the "Adaptive" category, we considered a model where $\beta$ is optimized so that the agent's policy complexity meets the capacity constraint $C$ (the "Adaptive: Capacity" model), and another where $\beta$ is optimized to target the agent's desired "reward aspiration" level (the "Adaptive: Value" model). One final model variant combines aspects of the first two adaptive models: the agent considers both capacity and aspiration levels when optimizing $\beta$ (the "Adaptive: Capacity-Value" model). See the Methods for a table comparing all model variants.

**Comparison models.**   Our policy compression models make the key assumption that human choice behavior and RT are sensitive to the marginal action probability and policy complexity. To test this claim, we compared our model to several comparison models that do not penalize policy complexity. First, we considered a "Standard RL" model of choice [3], as described in Eq 4. We also considered a version of the reinforcement learning working memory (RLWM) model that has been studied extensively by Collins and colleagues [15, 17, 20, 22, 23].

The RLWM model captures the parallel recruitment of working memory (WM) and reinforcement learning (RL) by simultaneously training two learning modules. It was originally developed to capture behavior in an instrumental learning task that examined the effects of memory load on learning and action selection. While the RLWM model successfully captures a wide range of behavioral effects, there is no mechanism for optimizing a trade-off between reward and policy complexity. We chose the RLWM model as a natural point of comparison because it makes explicit how memory constraints affect performance and response time in RL tasks like the ones we study here. Further details about the Standard RL and RLWM models can be found in the Methods.

In the current study, we used a simple instrumental learning task to show that human choice behavior conforms to the unique predictions of our policy compression model, and that models that do not penalize policy complexity cannot adequately capture our results.

## Experimental tasks

Using Amazon Mechanical Turk, we collected data from 200 subjects who participated in our online behavioral experiment. Subjects completed a series of three instrumental learning tasks that all shared the same experimental structure (Fig 3). Subjects were instructed to learn which of three key press responses was associated with a particular image stimulus to maximize reward. On each trial, subjects saw a single stimulus and were required to respond with a key press in under 2 seconds (with the exception of Task 3). Each stimulus was associated with one or more optimal (highest probability of reward) responses. After making a response, subjects were given feedback indicating whether their response was "correct" (a reward of +1 is earned) or "incorrect" (no reward is earned). The probability that the subject would receive "correct" feedback given their response was defined by a reward function that varied across tasks and conditions. If no response was made, the trial would be counted as "incorrect" and the next trial would begin. Subjects were told to maximize their payout, proportional to the number of "correct" responses made over the entire task. Each stimulus was presented 30 times in each task block (with the exception of Task 1).

Each task consisted of two block conditions, which we refer to as Q1 and Q2. Q1 always served as the "control" condition, whereas Q2 was designed to test a prediction of the policy compression model. In all tasks, we carefully designed Q1 and Q2 to have the same average and maximum reward values to control for motivational effects. In Task 1, both conditions shared the same reward function, but differed in their stimulus distribution (some stimuli

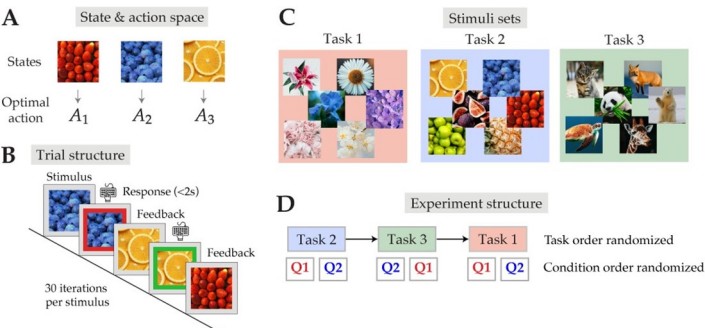

**Fig 3. Instrumental learning task. (A)** Example state-action space. Each task block comprised of three unique states (stimuli) and three possible actions. Each state was associated with one or more optimal action(s). **(B)** The experimental trial structure. Subjects made their response under 2 seconds and received negative (red) or positive (green) feedback in the form of a border around the image. Open source images from (A) and (B) were taken from: https://unsplash.com/photos/bunch-of-strawberries-KVv5lFOMY1E, https://unsplash.com/photos/sliced-orange-fruit-on-white-ceramic-plate-pCjw_ygKCv0, https://unsplash.com/photos/black-berries-lot-0DtoVEDaJbs **(C)** Each task used unique stimulus sets to prevent learning across tasks. Open source images were taken from: https://unsplash.com/photos/pink-flower-in-macro-lens-K3x_AkLVTAo, https://unsplash.com/photos/blue-flower-focus-photography-6ZyLeconAsg, https://unsplash.com/photos/purple-flowers-with-green-leaves-V4Pn7QeYdPQ, https://unsplash.com/photos/pink-and-white-flower-kkJuQhp9Kw0, https://unsplash.com/photos/white-cherry-blossom-in-close-up-photography-yRXuXvy4sQ4, https://unsplash.com/photos/white-daisy-in-bloom-during-daytime-3tYZjGSBwbk, https://unsplash.com/photos/bunch-of-strawberries-KVv5lFOMY1E, https://unsplash.com/photos/sliced-orange-fruit-on-white-ceramic-plate-pCjw_ygKCv0, https://unsplash.com/photos/black-berries-lot-0DtoVEDaJbs, https://unsplash.com/photos/closeup-photo-of-pineapple-5bdKZLqeySU, https://unsplash.com/photos/shallow-focus-photo-of-green-fruits-9Jl9Wk9juuE, https://unsplash.com/photos/a-bunch-of-figs-that-are-sitting-on-a-table--t52BM39yRs https://unsplash.com/photos/brown-turtle-swimming-underwater-L-2p8fapOA8, https://unsplash.com/photos/brown-fox-on-snow-field-xUUZcpQlqpM, https://unsplash.com/photos/selective-focus-photo-of-giraffe-D6TqIa-tWRY, https://unsplash.com/photos/panda-eating-bamboo-_9a-3NO5KJE, https://unsplash.com/photos/brown-tabby-kitten-sitting-on-floor-nKC772R_qog, https://unsplash.com/photos/polar-bear-on-snow-covered-ground-during-daytime-qQWV91TTBrE **(D)** Example experiment structure for one subject. The order of the three tasks, as well as the conditions within each task, were randomized across subjects.

appeared more frequently than others). In Task 2, the two conditions differed in their reward functions and the number of optimal responses per state. In Task 3, both conditions again shared the same reward function, but differed in the time limit within which subjects were required to make their response. Task order and block condition order within tasks were randomized across subjects. We encouraged independent learning of responses across stimuli by informing the subjects that multiple stimuli could share the same optimal response, or that a single state could have more than one optimal response. Finally, we ensured that subjects would not be biased towards any particular key on the keyboard by randomizing the mapping between stimuli and optimal responses, as well as the physical location of optimal responses in each task and condition. However, for the purpose of standardizing our data analysis, we re-mapped both stimuli and subjects' responses to be consistent with the depicted reward functions in each task figure.

**Model fitting and comparison.** We used maximum likelihood estimation to jointly fit choice and response time data for each subject. We fit one set of parameters per subject to capture their performance across all three tasks. In our quantitative model comparison, we found that many of the policy compression models scored very close in BIC (see S1 Fig for more details). As a result, we focus on qualitative model predictions to adjudicate between models. We simulated data from all candidate models and compared the dynamics of policy complexity, average reward, and response time during learning to determine the overall winning model. We also plotted the dynamic reward-complexity trade-off plot for each model, which shows on average, how subjects' policy complexity and reward evolve together over the course of each task. Lastly, we analyzed the correlation between RT and policy complexity (S2, S3 and S4 Figs).

Across all three tasks, the "Adaptive: Capacity-Value" model came closest to capturing the dynamics of learning in subjects' data in all three tasks. Though this model scored close to the "Adaptive: Capacity" and "Adaptive: Value" models in quantitative model comparison metrics, the other two adaptive models could not capture key aspects of the learning dynamics in Tasks 2 and 3 (for a more detailed explanation, please refer to S3 and S4 Figs, and their captions).

**State frequency biases action selection.** In Task 1, we asked whether an asymmetric state distribution, $P(s)$, would bias behavior in line with the predictions of our policy compression model (Fig 4). To do this, we varied the frequency of stimulus presentations in each condition. In Q1, all three stimuli were presented an equal number of times (30 presentations/stimulus, $P(s) = 0.33$ for all stimuli), while in Q2, one randomly chosen stimulus appeared three times more frequently than the others (90 presentations or $P(s) = 0.6$ for one stimulus, 30 presentations or $P(s) = 0.2$ for the other two). Therefore, for this task only, there are overall fewer trials in Q1 (90 trials) than in Q2 (150 trials). By ensuring that the low-frequency stimuli in Q2 were presented an equal number of times in both conditions, we enable a direct comparison of choice biases associated with those stimuli across conditions. Each stimulus had one unique optimal response that delivered the highest probability of reward (bolded), and two suboptimal responses that were equal in reward probability (green, orange, and purple boxes).

As a result of this stimulus frequency manipulation, the marginal distribution over actions, $P(a)$, should differ in Q1 and Q2 (Fig 4A). In Q1, all three actions should be chosen roughly equally, while in Q2, the optimal action $A_1$ should be overall chosen more frequently than the other two actions, simply because $S_1$ appears more often. The policy compression model, but not the Standard RL or RLWM models, predicts that this increased action frequency should bias action selection overall because of how the marginal action distribution enters into the optimal policy (Eq 3). As a result, subjects should show a preference for $A_1$ even in other states, and over other suboptimal actions with equal reward probability. This action preference should only be present in Q2 and not in Q1. Finally, the policy compression model predicts

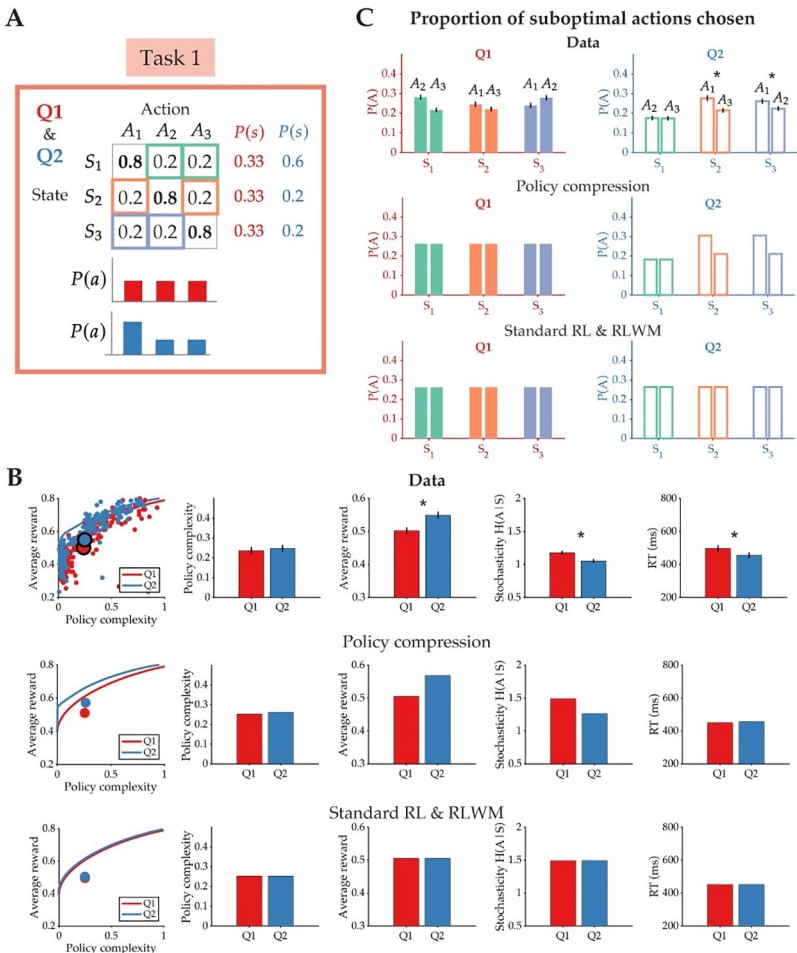

**Fig 4. Action selection is biased by the state distribution. (A)** Task 1 consisted of two conditions, Q1 and Q2, that shared the same reward function but differed in their state distribution (optimal actions for each state are in bold). As a result, the marginal action probability, $P(a)$, in Q2 is biased towards the optimal action of the state that appears most frequently (e.g., $A_1$ for $S_1$). The $P(a)$ depicted for each task condition is derived from the optimal unbounded policy, which assumes that $\beta \to \infty$ (i.e., assuming that subjects perfectly learn the reward function). **(B)** (Top) Policy complexity, average reward, stochasticity, and response time (RT) as a function of the two task conditions. (Middle) Qualitative behavioral predictions of the policy compression model. (Bottom) Qualitative behavioral predictions shared by the Standard RL and RLWM models. **(C)** The proportion of suboptimal actions chosen in each state. The marginal action probability resulting from the asymmetrical state distribution causes subject's behavior to be biased towards $A_1$ despite both suboptimal actions sharing the same expected reward value. The policy compression model alone predicts this action preference, and this bias does not appear for the suboptimal actions in condition Q1. All error bars indicate standard error.

overall higher expected reward values than the Standard RL and RLWM models because of how the relationship between policy complexity and reward changes with the state distribution (Fig 4B, second row; see also [6]). Therefore, the signature of policy compression in Task 1 is being able to earn more reward in Q2 than in Q1 with the same policy complexity. This reward advantage should also be greater for individuals with low policy complexity (as estimated from their task behavior), a hypothesis we explore in a subsequent section.

In Fig 4B, we show aggregated subject data and compare it to the qualitative predictions of each candidate model. Each data point on the reward-complexity trade-off plot in Fig 4B represents a single subject's performance in one task condition. (For more details about how

empirical policy complexity is calculated, see the Methods.) Policy complexity did not change between conditions [t(199) = -0.829, p = 0.408; Cohen's d = -0.059]. However, average reward earned was higher in Q2 [t(199) = -4.853, p<0.001; Cohen's d = -0.343], consistent with the unique predictions of the policy compression model. Stochasticity was significantly lower in Q2 [t(199) = 5.641, p<0.001; Cohen's d = 0.399], as well as response time [t(199) = 5.036, p<0.001; Cohen's d = 0.356].

As predicted, there was no systematic action preference in Q1, where both the state and marginal action distribution were uniform (Fig 4C, left). However, in Q2 subjects significantly preferred $A_1$ over the other suboptimal action in states $S_2$ and $S_3$, despite the probability of reward for both actions being equal [$\Delta P(A)$ for $S_2$: t(199) = 3.541, p<0.001; Cohen's d = 0.250 and $\Delta P(A)$ for $S_3$: t(199) = 2.182, p = 0.030; Cohen's d = 0.154], although the size of this effect is relatively small (Fig 4C, right). There was no difference in the proportion of suboptimal actions chosen in the high-frequency state $S_1$ [t(199) = 0.125, p = 0.900; Cohen's d = 0.009].

**Action frequency biases action selection.** In Task 2, we directly tested the prediction that the marginal action distribution, $P(a)$, biases subjects' behavior (Fig 5). To do this, we designed task conditions to vary in the number of shared optimal responses across stimuli. In Q1, each stimulus was associated with one unique optimal response that delivered deterministic reward, but in Q2, states $S_2$ and $S_3$ each had two optimal responses that both delivered deterministic reward (Fig 5A). Critically, one of these optimal responses, $A_1$, was shared across all three states. We predicted that in Q2, subjects would be more likely to choose the shared action over the other optimal action, despite both actions delivering deterministic reward. Additionally, we predicted that policy complexity would be lower and average reward higher in Q2 than in Q1, as the reward function in Q2 encourages policy compression via reliance on the marginal action distribution. This essentially means that in Q2, subjects can earn *more* reward than in Q1 with a *less* complex policy. We note here that compression can occur in two main ways: (1) earning more reward with the same policy complexity (as seen in Task 1), and (2) learning a simpler policy while earning the same, or more, reward. Both are valid ways to compress one's policy, as compression simply implies that people are ignoring some state information and taking advantage of the marginal action distribution.

In Fig 5B, we show aggregated subject data and compare it to the qualitative predictions of each model. As predicted, policy complexity was lower in Q2 [t(199) = 2.8213, p = 0.0052; Cohen's d = 0.199], yet average reward earned was higher [t(199) = -12.5759, p<0.0001; Cohen's d = -0.889]. Stochasticity was significantly lower in Q2 [t(199) = 9.8321, p<0.0001; Cohen's d = 0.695], as well as response time [t(199) = 7.1026, p<0.0001; Cohen's d = 0.502]. Note that for this task, the policy compression model makes similar qualitative predictions on most behavioral measures as the Standard RL and RLWM models. While the Standard RL and RLWM models are not constrained by capacity limits, (i.e., there is nothing in the models specifying what the complexity of the optimal policy should be) they can still match subjects' empirical policy complexity through parameter setting (as we demonstrated through the examples in Fig 2B–2D). So to enable a fair comparison, we simulated the comparison models to match the average policy complexity in our subject data. Though two different algorithms (e.g., Standard RL vs policy compression) can in theory generate the same policy complexity and reward, the pattern of their choice biases will differ significantly. This is exactly the case for Task 2, where the key behavioral signatures of policy compression are revealed when examining subjects' choice preferences in each condition (Fig 5C).

As predicted, there was no systematic action preference in Q1 where the marginal action distribution was uniform (Fig 5C, left). However, in Q2 subjects again significantly preferred $A_1$ over the other optimal action in states $S_2$ and $S_3$, despite the both actions delivering deterministic reward [$\Delta P(A)$ for $S_2$: t(199) = 4.350, p<0.001; Cohen's d = 0.308 and $\Delta P(A)$ for $S_3$: t

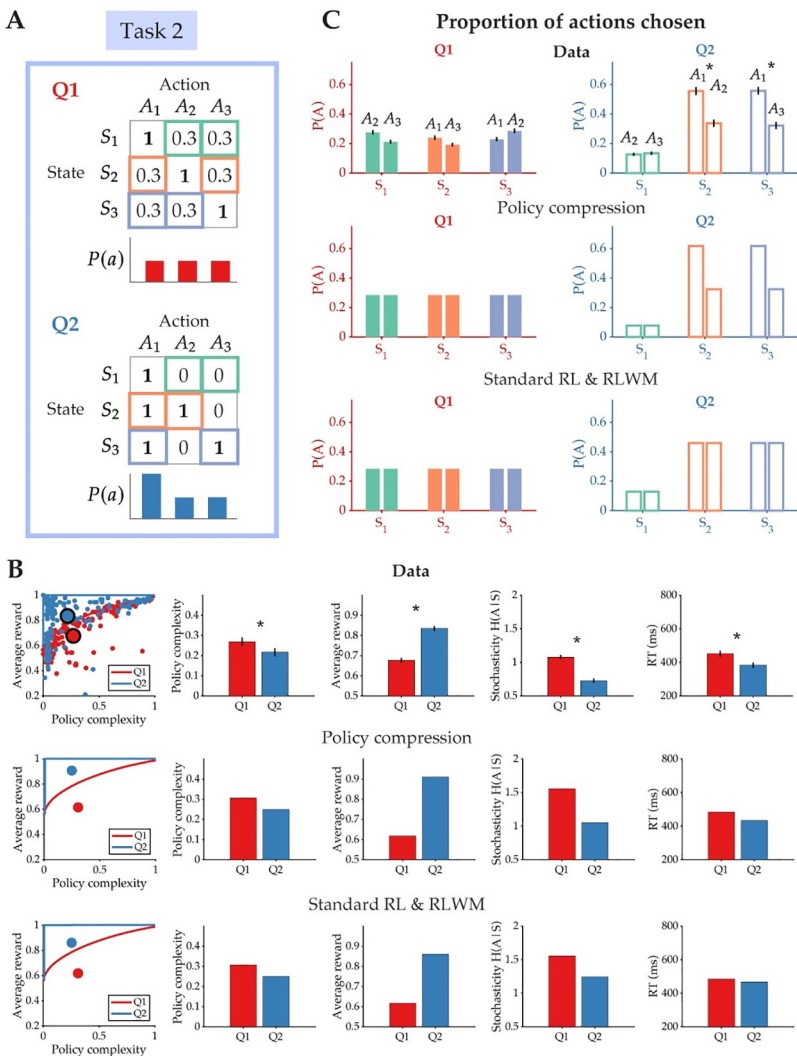

**Fig 5. Action selection is biased by the marginal action distribution. (A)** Task 2 consisted of two conditions, Q1 and Q2, that differed in the number of optimal actions per state (bolded). As a result, the marginal action probability, $P(a)$, in Q2 is biased towards the optimal action that is shared across all states (e.g., $A_1$). **(B)** (Top) Policy complexity, average reward, stochasticity, and response times (RT) as a function of the two task conditions. (Middle) Qualitative behavioral predictions of the policy compression model. (Bottom) Qualitative behavioral predictions shared by the Standard RL and RLWM models. **(C)** The proportion of actions with the same reward probability chosen in each state. The biased marginal action probability causes subjects to prefer $A_1$ over another optimal action that is equally rewarding. The policy compression model alone predicts this action preference. This biased preference does not appear for actions that share the same reward probability in condition Q1. All error bars indicate standard error.

$(199) = 4.763$, p<0.001; Cohen's d = 0.337] (Fig 5C, right). There was no difference in the proportion of suboptimal actions chosen in the state with only one optimal action, $S_1$ [t(199) = -0.698, p = 0.486; Cohen's d = -0.049]. This behavioral bias is a clear deviation from the predictions of the Standard RL and RLWM models, in which both optimal actions should be chosen equally. Notably, the size of this action bias is much larger in Task 2 than in Task 1: directly manipulating the action distribution produces a stronger action biases than attempting to manipulate it through state frequency.

**Fig 6. Bias scales with set size. (A)** An example of our analysis method from a condition where the set size, or number of states (nS), was 4. We averaged the proportion of suboptimal actions in states for which the optimal action was not high in marginal probability. **(B)** The average proportion of suboptimal actions that aligned with the high marginal probability ($A_1$), and those that did not ($A_2$) for each set size condition. **(C)** The difference between the proportion of suboptimal actions chosen as a function of set size. Error bars indicate standard error. Data source: [23].

We have focused our current analysis of action bias within one set size condition. However, a natural follow-up question is whether this bias increases as a function of set size. In previous work, we have shown that average policy complexity does not vary monotonically across set sizes, indicating a roughly constant resource constraint [10]. We interpreted this finding as consistent with the hypothesis that set size effects reflect the redistribution of a fixed resource across more states, resulting in lower precision per state [24]. Therefore, we should expect signatures of policy *compression* (in this case, a bias towards actions that are high in marginal probability) to increase with set size.

We confirmed this prediction by re-analyzing data from [23], which used a similar instrumental learning task to study learning across various set sizes with deterministic rewards (N = 40, Fig 6). By taking block conditions where optimal actions were shared across 2 or more states, we computed the difference between suboptimal actions in states for which the optimal action was not high in marginal probability (Fig 6A). In Fig 6B, we first show that across all set sizes, suboptimal actions that have high marginal probability are chosen more frequently over other suboptimal actions (p<0.001 for all set size conditions). We then show that the action bias, or the difference between the proportion of suboptimal actions chosen, $\Delta P(A) = P(A_1) - P(A_2)$, does increase slightly as a function of set size, though this increase is non-linear and possibly non-monotonic (Fig 6C). This may be due to averaging across various block conditions within one set size. For example, some blocks by design may have produced a stronger influence of the marginal probability on choice (i.e., optimal actions were shared across a majority of the states), while others may have produced less of an effect.

**Time pressure compresses policies.**   Now that we have shown that choice behavior aligns uniquely with the predictions of the policy compression model, we turn our attention to the hypothesis that actions are generated by time-dependent decoding. To perfectly decode an action from a state, the optimal policy complexity required is log $N$, where $N$ is the number of actions [11, 18]. In a Huffman code, the policy complexity corresponds to the number of bits that need to be inspected to reveal the coded action. If bits are inspected at a constant rate, response time should be a linear function of policy complexity, which can vary even when the number of states is held fixed (such as in our tasks).

We reasoned that time pressure should further reduce policy complexity in a capacity-limited agent by limiting the amount of time allowed for decoding actions from states. This is also the assumption that we built into our model linking trial-by-trial RT to policy cost. In Task 3, we tested this hypothesis by designing two task conditions that shared the same reward function but differed in the time allowed for subjects to make their response (Fig 7A). In Q1, subjects were allowed 2 seconds to make their key press response (as in the other two tasks), while in Q2, they were only given 1 second. If subjects failed to respond within the 1 second time window, they were shown a warning message that encouraged them to respond faster or risk

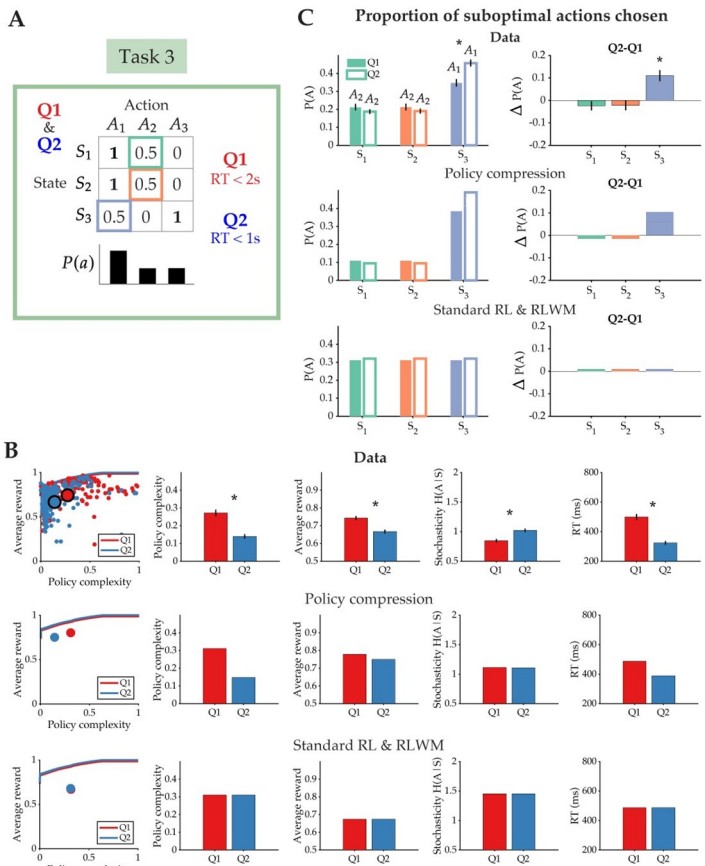

**Fig 7. Policies are more compressed under time pressure. (A)** Task 3 consisted of two conditions, Q1 and Q2, that shared the same reward function (optimal actions in bold) but differed in the time allowed for subjects to make their response. In this task, the marginal action probability, $P(a)$, is the same for both conditions, and is biased towards one action, $A_1$. **(B)** (Top) Policy complexity, average reward, stochasticity, and response times (RT) as a function of the two task conditions. (Middle) Qualitative behavioral predictions of the policy compression model. (Bottom) Qualitative behavioral predictions shared by the Standard RL and RLWM models. **(C)** The proportion of suboptimal actions chosen in each state. Under time pressure (condition Q2), there is a greater influence of the marginal action probability on choice behavior. Subjects choose suboptimal action $A_1$ in $S_3$ more often than they did when given more time to respond. The policy compression model alone predicts this action preference. All error bars indicate standard error.

having their bonus for the task withheld. Importantly, the marginal action probability was concentrated on one action, as two out of the three states shared an optimal action that delivered deterministic reward. We predicted that under time pressure (Q2), subjects would further compress their policies, reducing policy complexity and choosing suboptimal actions more often (Fig 7A, boxed. Here, we define suboptimal action as the action with the second highest reward probability). In particular, we predicted that subjects would be biased to choose $A_1$ in $S_3$ more often than when not under time pressure (purple box), but that there would be no difference in the expression of suboptimal action $A_2$ in $S_1$ and $S_2$ (green and orange boxes) across conditions. This is because the marginal action distribution is concentrated on $A_1$ rather than $A_2$.

In Fig 7B, we show aggregated subject data and compare it to the qualitative predictions of each model. As predicted, policy complexity was significantly lower in the time pressure

condition (Q2) [$t(199) = 7.082$, $p<0.001$; Cohen's $d = 0.501$], as well as average reward [$t(199) = 6.948$, $p<0.001$; Cohen's $d = 0.491$]. Stochasticity significantly increased under time pressure [$t(199) = -5.536$, $p<0.001$; Cohen's $d = -0.391$], and response time decreased as expected to stay within the new time constraint [$t(199) = 12.4826$, $p<0.001$; Cohen's $d = 0.883$]. In the Standard RL and RLWM models, there is no mechanism for how time pressure should change subjects' policies, and therefore no predicted qualitative differences between Q1 and Q2.

As predicted by the policy compression model, there was no difference in the proportion of suboptimal actions chosen between conditions in $S_1$ and $S_2$ [$\Delta P(A)$ for $S_1$: $t(199) = -1.181$, $p = 0.239$; Cohen's $d = -0.0835$ and $\Delta P(A)$ for $S_2$: $t(199) = -1.061$, $p = 0.290$; Cohen's $d = -0.075$] (Fig 7C). However, in Q2 subjects were biased to choose $A_1$ in $S_3$ more often than in Q1, indicating greater policy compression and an increased reliance on the marginal action probability when under time pressure. [$\Delta P(A)$ for $S_3$: $t(199) = 4.488$, $p<0.001$; Cohen's $d = 0.317$]. This increased influence of the marginal action distribution is not predicted by the Standard RL and RLWM models, in which suboptimal actions should be chosen equally across conditions in all states, regardless of time limits on response.

## Individual differences in policy complexity predict action bias and earned reward

We now summarize and follow up the key results from our within-subject analyses with between-subject analyses that highlight systematic differences in individuals' behavior across all three tasks. Specifically, we analyze how subjects' empirical policy complexity relates to their choice bias in each task, and how this bias influences earned average reward.

First, the state distribution affects the relationship between reward and policy complexity in the compression framework. An asymmetrical state distribution makes it possible for subjects to earn more reward with the same policy complexity (Fig 4B, middle), and causes subjects to be biased towards suboptimal actions with high marginal probability. This pattern of choice bias is distinct from a Standard RL or RLWM strategy where action selection in each state is treated independently. Subjects' average action bias (i.e., the bias towards choosing a suboptimal action with high marginal probability over another suboptimal action with the same reward probability) in Q2 decreased as a function of policy complexity, although this effect is small [Pearson's correlation: $r = -0.179$, $p = 0.011$] (Fig 8A).

Furthermore, we reasoned that subjects with low complexity (defined as the empirical policy complexity estimated from choice behavior in each task condition) would benefit more from action bias, since it is more efficient to focus one's limited cognitive resources on states (and their optimal actions) that appear most frequently. In line with this prediction, we found that subjects' increase in average reward from Q1 to Q2 was positively correlated with action bias for subjects with low [Pearson's correlation: $r = 0.322$, $p<0.001$], but not high [Pearson's correlation: $r = -0.112$, $p = 0.313$], policy complexity Fig 8B). In other words, subjects with low complexity policies are able to take advantage of their action bias to earn more reward via policy compression, though they did not necessarily increase their average reward more than those with high complexity policies [$t(198) = -1.278$, $p = 0.203$; Cohen's $d = -0.178$].

Second, this perseverative action bias is even greater when there is redundancy in the reward-maximizing policy across states. Even when there was more than one optimal action per state, subjects consistently preferred the optimal action with a higher marginal action probability. This action bias was, again, more pronounced for individuals with low policy complexity: we found a strong negative relationship between average action bias and policy complexity [Pearson's correlation: $r = -0.817$, $p<0.001$] (Fig 8C). While action bias and policy complexity are estimated from the same behavioral data, their relationship is not entirely

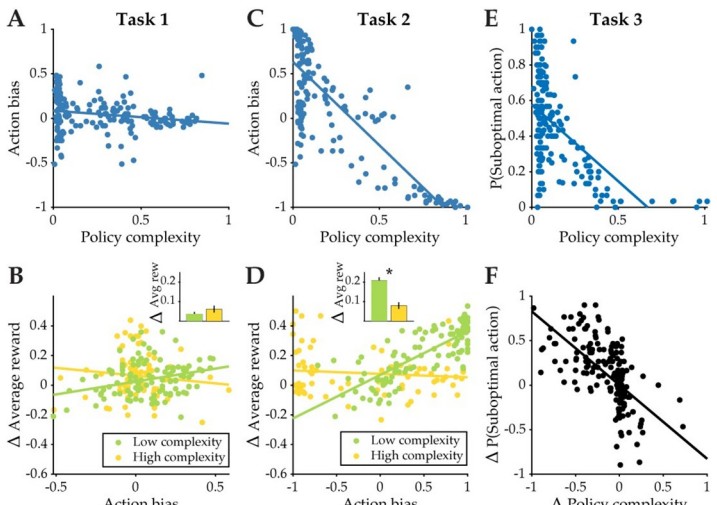

**Fig 8. Action bias decreases with policy complexity and increases reward for subjects with low policy complexity.**
**(A)** Action bias as a function of policy complexity in Task 1. Each data point represents a single subject's mean action bias, or the bias towards choosing a suboptimal action with high marginal probability over another suboptimal action with the same reward value), plotted against their policy complexity (data from Q2 only). **(B)** The change in average reward across task conditions as a function of subjects' action bias for subjects with low and high complexity policies (mean split of data). Each data point represents a single subject's difference in earned average reward across conditions (Q2-Q1), plotted against their mean action bias in Q2. **(C)** Same as (A) but for Task 2. **(D)** Same as (B) but for Task 2. **(E)** The proportion of choosing the suboptimal action in $S_3$ as a function of policy complexity in Task 3 (data from Q2 only). **(F)** The change in proportion of choosing the suboptimal action (Q2-Q1) as a function of the change in policy complexity (Q2-Q1) due to time pressure in Task 3.

tautological: though it is unlikely for a high complexity policy to produce high action bias, it is possible to have a low complexity policy (e.g., taking random actions regardless of state) without being biased towards the action with highest marginal probability. Our analysis here shows that the relationship between action bias and complexity becomes more pronounced by directly manipulating the action distribution (as in Task 2), rather than the state distribution (as in Task 1). Consistent with Task 1, we also found that the change in average reward earned is correlated with greater action bias in subjects with low [Pearson's correlation: r = 0.793, p<0.001], but not high [Pearson's correlation: r = -0.108, p = 0.345], policy complexity (Fig 8D). Subjects with low complexity policies also had a greater increase in earned average reward across conditions than those with high complexity policies [t(198) = 5.354, p<0.001; Cohen's d = 0.788], indicating that their action bias allowed them to significantly increase reward earnings despite their resource limit.

Third, we provided evidence for a strong relationship between time pressure and policy compression: under a tight time constraint, subjects reduce their policy complexity and sacrifice reward. The relationship between response time and policy complexity is not only present in Task 3; indeed, there is a strong positive correlation between RT and complexity in all three tasks [Task 1: Pearson's correlation: r = 0.513 for Q1, r = 0.609 for Q2, both p<0.001; Task 2: Pearson's correlation: r = 0.591 for Q1, r = 0.669 for Q2, both p<0.001; Task 3: Pearson's correlation: r = 0.638 for Q1, r = 0.563 for Q2, both p<0.001], a feature that can only be replicated by the policy compression models (S2, S3 and S4 Figs). On the other hand, both the Standard RL and RLWM models predict a negative relationship between policy complexity and RT, which is a clear deviation from the data (see panels E and F of S2, S3 and S4 Figs). Under time pressure, subjects—especially those with low policy complexity—rely more on their action

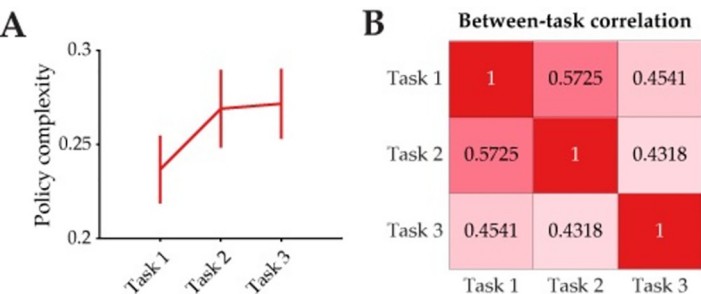

**Fig 9. Policy complexity is correlated across tasks.** (A) Average policy complexity in the control condition (Q1) for each task. Error bars indicate standard error. (B) Moderate between-task correlations in policy complexity.

history to make choices: Fig 8E shows a negative relationship between the tendency to choose a suboptimal action with high marginal probability and policy complexity. Additionally, subjects' decrease in policy complexity under time pressure was correlated with an increase in choosing the suboptimal action across conditions [Pearson's correlation: r = -0.630, p<0.001] (Fig 8F).

Finally, we asked whether policy complexity reflected a fixed resource that remained roughly constant across tasks, or whether it varied across tasks. On average, subjects' policy complexity in the control condition (Q1) remained roughly constant across tasks (Fig 9A) [one-way ANOVA: $F_{(2,597)} = 1.032$, p = 0.357]. Additionally, we found a moderate positive correlation between policy complexities across tasks (Fig 9B) [Pearson's correlation coefficients ranging from 0.432 to 0.573, all p<0.001], which shows that subjects used roughly the same amount of cognitive resources for each task. Subjects' empirical policy complexity may reflect an individual's actual cognitive capacity limit (in a trait-like fashion), or simply reflect the amount of cognitive effort that the individual has chosen to allocate for the experimental tasks. While our experiment was not designed to tease apart these two possibilities, future work may consider investigating how independent, trait-like measures such as working memory capacity could predict individuals' policy complexity in experimental tasks. In either case, we know from our results that policy complexity is flexible to some degree within an individual, as it can be shaped by the task structure itself and by exogenous factors such as time pressure and memory load.

## Discussion

In this paper, we tested the hypothesis that humans prefer simpler policies, as predicted by a capacity-limited reinforcement learning model. Across three tasks, we found that human subjects utilize structure in the relationship between states, actions, and rewards to "compress" their policies. This strategy allows subjects to discard some state information (i.e., reduce their policy complexity) without sacrificing reward. As a consequence of policy compression, people are systematically biased towards actions they have chosen most frequently in the past. This bias persists even when multiple optimal actions are available, and increases under both time pressure and memory load. These results are uniquely explained by models that balance between two computational goals: reward maximization and policy compression under a capacity limit.

We found that the "Adaptive: Capacity-Value" model best described our data on both quantitative and qualitative measures. This model assumes that the dynamics of learning are driven both by the agent's capacity limit as well as a desired aspiration level. We found that

this method of optimizing the policy allowed agents to flexibly adapt to a variety of environments with different reward-complexity trade-off landscapes, such as in our different task conditions. Recall that the policy compression theory assumes that learned policies have complexities that equal an agent's capacity limit. However, there may be situations where agents choose instead to "satisfice" at some aspiration level (for example, an agent might be content with an average reward value of 0.8) and not make use of all their computational resources. In fact, previous work has shown sensitivity of human decision making to aspiration level, in simple as well as in more complex tasks such as financial investing [25–27]. Future models should consider the effects of a desired aspiration level on the dynamics of learning and decision making.

Our study is the first designed to directly test the unique behavioral predictions of the policy compression framework, which has already enjoyed success in explaining a range of behavioral phenomena. A key distinguishing feature of our model from others that consider both RL and memory capacity (such as the RLWM model [15]) is the application of rate-distortion theory [4, 28] to reinforcement learning to characterize decision-making under a capacity limit. This framework allows us to derive the form of an optimal policy and the accompanying process model that optimizes the trade-off between reward and policy complexity [5–7, 11, 12].

Our modeling framework allows us to interpret well-studied phenomena through a new, normative lens. For instance, though many previous studies have examined the influence of time constraints on choice behavior [29, 30], policy compression offers a normative rationale for the relationship between response time and policy complexity. Time pressure reduces an agent's capacity limit, shortening the expected code length that determines how long it takes to decode actions from state representations. This, in turn, leads to an even greater bias towards previous actions according to the optimal form of the policy. We note one caveat: in this study we did not explicitly disentangle policy learning from its implementation, which raises the question of whether reduced complexity under time pressure is a feature of time-dependent *decoding* alone, or whether it a priori affects policy *encoding* as well. However, since previous work [11, 31] has shown that policy compressibility relates to its learnability (i.e., simpler or more compressed policies are easier to learn), it is reasonable to assume that time pressure also causes agents to learn overall simpler policies.

Our framework also incorporates ideas previously proposed in models combining response time and choice. For example, [20] reasoned that the uncertainty over actions prior to encoding the current stimulus should affect decision time, a term that they added into their joint model of RT and choice. This prior uncertainty enters into choice by scaling down the drift rates of each action equally. If one action is used more frequently than the others, the prior uncertainty over actions is smaller, and drift rates are faster. While our model shares the similar approach of considering how prior information influences aspects of action selection, it differs in that the prior distribution over actions affects not only RT but the choice itself (which is why the RLWM model cannot capture our result).

While our study provides compelling evidence for several key predictions of the policy compression model, it also has several limitations. First, while our modeling procedure was able to distinguish between models that do and do not penalize policy complexity, it was unable to unambiguously identify the correct model variant within the class of policy compression models. Designing specific experiments to distinguish between fixed and adaptive policy compression models is a potential area for further research. Second, we relied on a previously published dataset to test the prediction that compression increases with memory load. While we found a relationship between set size and action bias, the dataset used was not explicitly designed to investigate whether bias increases monotonically with set size, or if there is indeed a "plateau" effect for higher memory loads. Exploring how memory load influences

compression and perseverative action biases remains an avenue for future investigation. Finally, we note that a majority of subjects' behavior deviated from the optimal reward-complexity trade-off curve, which indicates additional sources of error and bias that are not due to the marginal action probability. For example, it is certainly possible that other forms of perseveration (e.g., repeating actions that do not have high marginal probability, such as in sequential effects or motor habits) may co-occur with those that arise from capacity limits, but their origins are beyond the scope of this paper. Future experiments could be uniquely designed to tease apart different sources of perseveration-like effects.

A limitation of our theoretical framework is that it assumes people always prefer simpler policies (all other things being equal), whereas some evidence suggests a preference for *empowerment* (i.e., more complex policies) in certain situations, such as games without an explicit reward function [32]. In such situations, policy complexity itself can become a source of intrinsic motivation. Intuitively, an agent with high policy complexity has more control over the environment, which is useful for carrying out many different tasks. An important challenge for future work will be to disentangle the conditions in which people seek or avoid policy complexity.

This study adds to a larger body of research that focuses on how agents can utilize environmental structure to compress or simplify behavior, which may facilitate generalization in novel situations [33, 34]. Understanding the relationship between policy compression and generalization to new tasks is an interesting direction for future research, with potential implications for designing artificial learning agents with human-like inductive biases. Our study also highlights an important distinction between policy complexity (the number of bits needed to encode and decode a policy), statistical complexity (the amount of data needed to learn a policy), and computational complexity (the number of operations needed to execute an algorithm), which is important for placing our model in the broader context of other common RL algorithms such as model-free and model-based RL. For example, while model-based RL is more computationally complex than model-free RL (as it requires more operations to implement), it is not necessarily more statistically complex [31], and also does not necessarily require a more complex policy. Since model-free and model-based RL both require attention to the state for selecting actions, the complexity of the policies learned with either algorithm will vary depending on the particular state space and reward structure that learning takes place in. A more comprehensive theoretical analysis of how policy complexity differs in common RL algorithms would be an interesting area for further exploration.

While our finding that people are biased towards past choices is not new, our study suggests ways to accommodate or leverage these biases. Understanding how individuals behave under cognitive constraints can inform the creation of decision environments that align with these behavioral tendencies, promoting more effective decision making. For example, consider our result from Task 1, where individuals with lower complexity benefited the most from the asymmetric state distribution. The "design" of this choice environment enabled them to leverage their biases to earn more reward, compared to an environment with a uniform state distribution. In the same vein, knowing how people adapt their choice behavior under time pressure can shape the way information is presented to busy, time-poor individuals facing important decisions.

These ideas fall under the umbrella of "libertarian paternalism," the philosophy that societal structures and policies can be thoughtfully designed to positively influence people's choices [35]. For instance, the "choice architecture" of decision environments, such as default options, can be strategically selected to impact group or individual decision-making. Examples include automatic enrollment in retirement savings plans [36] or setting renewable energy sources as the default option [37], both of which have been shown to positively influence people's choices.

These default settings take advantage of people's perseverative biases, especially when they don't have the time or cognitive resources to properly evaluate their options before deciding. Some have even suggested that these choice environments should be "engineered" by using quantitative models such as ours to shape choice behavior [38]. We believe that computational models of policy compression (and its accompanying biases) may be powerful tools for choice architecture design.

## Methods

### Theoretical framework

The theoretical framework of policy optimization under an information-theoretic capacity limit was originally developed in several papers by Tishby and his collaborators [5–7]. Still and Precup [8] and Lerch et al. [9] developed online RL algorithms to learn a policy that optimizes the trade-off between reward and policy complexity through interactions with the environment. We further built upon these ideas [10, 11] by introducing a process model that (1) incrementally modifies the policy based on reward feedback that directly penalizes policy complexity, and (2) specifies how an agent's reward-complexity trade-off should evolve during learning. In the current study, we (3) built trial-by-trial response time predictions into our process model, which allows us to directly test the hypothesis that RT in part reflects the time-sensitive decoding of a compressed code. In what follows, we review the general framework along with the process model.

**Policy compression via capacity-limited reward optimization.** We assume that the optimal policy for an unbounded agent maximizes expected reward:

$$\pi^* = \underset{\pi}{\mathrm{argmax}}\ V^\pi, \tag{7}$$

where $V^\pi$ is the expected reward under policy $\pi$:

$$V^\pi = \sum_s P(s) \sum_a \pi(a|s) Q(s, a). \tag{8}$$

Here $P(s)$ is the probability of state $s$, and $Q(s, a)$ is the expected reward in state $s$ after taking action $a$.

A capacity-limited agent faces the additional constraint that its policy complexity (information rate) cannot exceed its capacity $C$. Behavioral evidence shows that people are subject to a capacity limit even in simple instrumental learning tasks [12, 15]. Policy complexity is formally defined as the mutual information between states and actions, which measures the average number of bits necessary to encode a policy:

$$I^\pi(S; A) = \sum_s P(s) \sum_a \pi(a|s) \log \frac{\pi(a|s)}{P(a)}, \tag{9}$$

where $P(a) = \sum_s P(s)\pi(a|s)$ is the marginal probability of choosing action $a$. Policy complexity is higher when the policy depends strongly on the state: it is maximized when each state maps to a unique action, and it is minimized when the distribution over actions is the same in each state (Fig 1B).

A capacity-limited agent is faced with the optimization problem of maximizing expected reward subject to its capacity limit, $C$:

$$\underset{\pi}{\mathrm{argmax}} \quad V^{\pi}$$
$$\text{subject to} \quad I^{\pi}(S; A) = C. \tag{10}$$

Two other necessary constraints ($P(a)$ must be non-negative and sum to 1) are left implicit.

Another way to view the same problem is to minimize policy complexity subject to a fixed aspiration level $R$ (desired reward rate; see [8]).

$$\underset{\pi}{\mathrm{argmin}} \quad I^{\pi}(S; A)$$
$$\text{subject to} \quad V^{\pi} = R. \tag{11}$$

The two optimization problems can lead to the same optimal policy if the aspiration level $R$ is chosen to be the highest expected reward achievable under capacity $C$ (Fig 1D). Both constrained optimization problems can be equivalently expressed and solved in a Lagrangian form:

$$\pi^* = \underset{\pi}{\mathrm{argmax}}\, \beta V^{\pi} - I^{\pi}(S; A) + \sum_s \lambda(s)(\sum_a \pi(a|s) - 1), \tag{12}$$

with Lagrange multipliers $\beta \geq 0$ and $\lambda(s) \geq 0$ (the 3rd term ensures proper normalization, and we will leave it implicit in subsequent equations). Solving Eq 12 leads to the optimal policy, $\pi^*$ [5, 6, 8]:

$$\pi^*(a|s) \propto \exp[\beta Q(s, a) + \log P^*(a)], \tag{13}$$

which is a softmax function with an added term $P^*(a) = \sum_s P(s)\pi^*(a|s)$ that biases the policy towards actions that are chosen frequently across all states. The Lagrange multiplier $\beta$ acts as the familiar "inverse temperature" parameter that regulates the exploration-exploitation trade-off via the amount of stochasticity in the policy [3]. It also indexes how state-dependent a policy is: When $\beta$ is close to 0, the policy will be state-independent, driven by actions that are overall chosen more frequently (the $P^*(a)$ term, Fig 2B). As $\beta$ increases, the policy will select actions that yield the most reward, conditional on the current state (the $Q(s, a)$ term, Fig 2D). The policy also becomes more state-dependent with increasing $\beta$, thus increasing policy complexity. Finally, $\beta$ is also implicitly related to the capacity constraint—its inverse is the slope of the reward-complexity trade-off curve evaluated at the capacity constraint $I(S; A) = C$:

$$\beta^{-1} = \frac{dV}{dI(S; A)}. \tag{14}$$

In other words, at each value of $C$ there exists a unique $\beta$ that constitutes one point on the optimal reward-complexity trade-off, and thus the entire trade-off curve is constructed by evaluating Eq 10 at different values of $C$ (Fig 1D). In general, there is no analytical form for the mapping from $C$ to $\beta$, which means that an agent with access to its capacity may not be able to specify the inverse temperature corresponding to the optimal policy. In previous work [12], we have used a variant of the Blahut-Arimoto algorithm [39] to find the optimal policy. The algorithm iterates between updating $\pi(a|s)$ according to Eq 13 and updating $P(a)$ under the current policy. By performing this optimization for a range of $\beta$ values, we can identify the point on the reward-complexity curve that characterizes the optimal policy for a given capacity constraint, $C$ or a fixed attainable reward, $R$.

 

**A process model for learning under constraints.** The Blahut-Arimoto algorithm requires direct knowledge of the state-action value function and is computationally intractable when the state space is large (because it requires marginalization over all states). We therefore derived a tractable process model based on an "actor-critic" architecture from reinforcement learning (RL) (see also [10, 11]). In the following sections, we also expand upon our previous work by bridging the process model to specific response time (RT) predictions, providing a direct theoretical link for how policy cost should impact RT behavior.

We can cast the optimization problems in Eqs 10 and 11 in a form amenable to RL by rewriting the Lagrangian in Eq 12 (dropping the normalization term for simplicity):

$$\pi^* = \underset{\pi}{\operatorname{argmax}} \, \mathbb{E}\left[\beta r - \log \frac{\pi(a|s)}{P(a)}\right]. \tag{15}$$

To find the optimal policy $\pi^*$, the cost-sensitive agent must find the policy parameters $\theta^*$ that maximize expected reward relative to the policy complexity cost. Throughout this paper, we will use the term policy *cost* to refer to $\log \frac{\pi(a|s)}{P(a)}$, and policy *complexity* to refer to its expectation, $I^\pi(S; A)$. The policy cost indexes the cost of taking a specific action $a$ in the state $s$ by quantifying the deviation of the current, state-specific action policy $\pi(a|s)$ from the marginal action probability $P(a)$. In this way, the policy cost captures the amount of state-specific information used to select a particular action in a single trial or time step. Note that the policy cost corresponds to a form of "entropy regularization" [40] in the special case where $P(a)$ is uniform. Allowing a non-uniform marginal is an important feature for capturing some of our experimental results.

We define the space of policies by adopting the following functional form:

$$\pi_\theta(a|s) \propto \exp[\beta\theta_{sa} + \log P(a)], \tag{16}$$

where $\theta_{sa}$ can be understood as an action "propensity" (the degree to which action $a$ tends to be selected in state $s$). By modifying the policy parameters $\theta$ to follow the gradient of Eq 15, we obtain a "policy gradient" algorithm [3]:

$$\Delta\theta = \begin{cases} \alpha_\theta \delta[1 - \pi_\theta(a|s)]\beta & \text{for the chosen action} \\ -\alpha_\theta \delta \pi_\theta(a|s)\beta & \text{for unchosen actions} \end{cases} \tag{17}$$

where $\alpha_\theta$ is the "actor" (policy) learning rate and

$$\delta = \beta r - \log \frac{\pi_\theta(a|s)}{P(a)} - \hat{V}(s), \tag{18}$$

is the prediction error of the "critic" $\hat{V}(s)$, which is updated according to:

$$\Delta\hat{V}(s) = \alpha_V \delta, \tag{19}$$

where $\alpha_V$ is a learning rate. At this point, the reader may wonder why the policy parameters are updated according to the policy gradient rather than simply treated as state-action Q-values and updated directly based on reward, as suggested by the form of the optimal resource-constrained policy. While this might be more straightforward for small state and action spaces where we can use look-up tables, applications of this framework to high-dimensional or continuous state and action spaces create challenges for value function approximators. In contrast, the optimal policy may be simpler to approximate with a relatively small number of parameters [3]. Indeed, precisely because we are regularizing towards simpler policies, we expect this to be typically true.

 

We incrementally estimate the marginal action probabilities with an exponential moving average:

$$\Delta P(a) = \alpha_P[\pi_\theta(a|s) - P(a)], \tag{20}$$

with learning rate $\alpha_P$.

Finally, the trade-off parameter $\beta$ can either be fixed (we called this the "Fixed" model) or adaptively optimized through learning. This can be done in several ways. First, $\beta$ can be optimized so that policy complexity meets the capacity constraint, $C$:

$$\Delta\beta = \alpha_\beta(C - \xi), \tag{21}$$

where $\xi$ is the agent's estimate of its own policy complexity, updated with an exponential moving average:

$$\Delta\xi = \alpha_\xi\left[\log\frac{\pi_\theta(a|s)}{P(a)} - \xi\right], \tag{22}$$

with learning rate $\alpha_\xi$ (we fixed $\alpha_\xi = 0.01$). We called this the "Adaptive: Capacity" model.

The second way to adaptively optimize $\beta$ is to target a desired "reward aspiration" level, $R$:

$$\Delta\beta = \alpha_\beta(R - \rho), \tag{23}$$

where $\rho$ is the agent's current estimate of the average reward, also updated via moving average:

$$\Delta\rho = \alpha_\rho(r - \rho), \tag{24}$$

with learning rate $\alpha_\rho$ (we fixed $\alpha_\rho = 0.01$). We called this the "Adaptive: Value" model.

Finally, we considered a third, hybrid model which combines elements of the first two adaptive models. In this "Adaptive: Capacity-Value" model, the agent considers both capacity and aspiration levels when adaptively optimizing $\beta$:

$$\Delta\beta = \alpha_\beta\left(\frac{C - \xi}{R - \rho} - \beta\right). \tag{25}$$

This model variant adapts beta towards an approximation of the inverse slope (i.e., $\frac{C-\xi}{R-\rho}$) at a point on the optimal reward-complexity trade-off curve, and allows the agent to flexibly adapt to a variety of environments with different trade-off landscapes. For example, when the current policy complexity estimate $\xi$ deviates from the capacity constraint $C$ more than the current reward $\rho$ deviates from the aspiration level $R$ (i.e., the numerator is larger), $\beta$ should be updated towards a value greater than 1, and the policy should become more complex, or state-dependent (S5 Fig, left). But when the deviation between current reward and aspiration level is greater than the deviation between policy complexity and capacity (i.e., the denominator is larger), the current policy is suboptimal and lies below the optimal trade-off curve (S5 Fig, right). In this case, $\beta$ should be updated towards a value less than 1 to decrease complexity and move closer to the optimal reward- complexity trade-off. As a result, the agent may end up learning a policy that does not utilize its full capacity $C$, but that still optimally maximizes reward for the chosen policy complexity. In practice, we can not know what value of $\beta$ subjects use at the start of learning, so we initialize $\beta$ to a fitted value between 1 and 10 and allow it to adapt via Eq 25.

**Comparison models.** We compared our cost-sensitive models to several comparison models that do not penalize policy complexity. First, we consider a "Standard RL" model of choice [3], where an agent learns action-values for each state, $Q(s, a)$, by updating it's estimate

on each trial using a delta rule [41]:

$$\Delta Q(s, a) = \alpha_Q \delta, \tag{26}$$

where

$$\delta = r - Q(s, a) \tag{27}$$

is the reward prediction error, $\alpha_Q$ is the learning rate, and $r$ is the reward received on the current trial after taking action $a$ in state $s$. These state-action values are then transformed into choice probabilities via a softmax function:

$$\pi(a|s) \propto \exp[\beta Q(s, a)], \tag{28}$$

where $\beta$ is the inverse temperature parameter.

As mentioned in the main text, we also considered a version of the reinforcement learning working memory (RLWM) model, studied extensively by Collins and colleagues [15, 17, 20, 22, 23]. In particular, we implement the model described in [20] which built on the original RLWM model by adding in response time predictions. This makes it a natural comparison because like our model, it makes specific predictions about how memory constraints affect performance and response time. The RL module is characterized by Eqs 26 and 27. The WM module learns stimulus-response associations $W(s, a)$ with a fixed learning rate $\alpha_{WM} = 1$:

$$\Delta W(s, a) = \alpha_{WM}[r - W(s, a)], \tag{29}$$

which means that it has the advantage of perfect learning of the observed outcome, in contrast to a gradual RL process. However, because working memory is vulnerable to short-term forgetting, the WM module also includes trial-by-trial decay of $W$:

$$\Delta W = \phi(W_0 - W) \tag{30}$$

where $\phi$ represents time-based decay or forgetting of items held in short-term memory. Practically, $\phi$ is a fitted parameter that draws W (over all stimuli and actions) toward their initial values $W_0 = \frac{1}{n_A}$, and $n_A$ is the number of actions. Additionally, to capture the asymmetrical effects of learning from positive and negative feedback, the learning rates in Eqs 26 and 29 are scaled whenever the agent receives "incorrect" feedback on a given trial:

$$\alpha = \gamma \alpha \tag{31}$$

where $\gamma$ controls the degree of perseveration (with lower values causing more perseveration).

The WM and RL policies ($\pi_{RL}$ and $\pi_{WM}$) are computed using the respective softmax functions:

$$\pi_{RL}(a|s) \propto \exp[\beta_{RL} Q(s, a)] \quad \pi_{WM}(a|s) \propto \exp[\beta_{WM} W(s, a)]. \tag{32}$$

We set both $\beta_{RL}$ and $\beta_{WM}$ to 50. The two policies are then combined in the final policy $\pi$ via a weighted sum:

$$\pi(a|s) = w\pi_{WM}(a|s) + (1 - w)\pi_{RL}(a|s), \tag{33}$$

where $w$ represents the contribution of WM to choice behavior and is itself modulated by two additional parameters, the working memory capacity $C$, and the initial WM weighting $\rho$:

$$w = \rho \cdot \min\left(1, \frac{C}{n_S}\right), \tag{34}$$

where $n_S$ is the set size, or number of unique stimuli (in our study, set size is always fixed at $n_S = 3$).

Note that while both the policy compression and RLWM models have a capacity parameter $C$, the interpretation is slightly different. In the policy compression model, $C$ defines an upper bound on mutual information, while in the RLWM model, $C$ is the number of items that can be held in working memory. Therefore, if the set size exceeds the capacity $C$, the influence of WM on action selection is reduced.

To build RT predictions into both the "Standard RL" and "RLWM" models, we borrow from [20] who used an evidence accumulation model, the Linear Ballistic Accumulator (LBA), to link choice probabilities to trial-by-trial response times. Specifically, there are individual evidence accumulators for each action that "race" and terminate at an upper bound $B$. The accumulator that reaches the bound first is the action that is executed, and the time-to-bound is the response time (RT). In the basic LBA model, the mean drift rate $v_i$ of each accumulator $i$ represents the evidence accumulation process of different competing actions. On each trial, $v_i$ is scaled proportionally by its associated action probability from the policy $\pi$:

$$v_i = \eta \pi(a_i|s), \tag{35}$$

where $\eta$ is a scaling parameter. This model of RT is consistent with assumptions from the actor-critic framework, where state-action weights in the striatum govern decision latency. Similar to our RT model, [20] additionally assume that prior uncertainty over actions would influence decision time. This prior uncertainty term $H_{\text{prior}}$ was modeled by computing an average policy $\vec{\pi}^\mu$ that averages action weights for each action over each state and across all states:

$$\vec{\pi}^\mu = \frac{1}{n_S} \sum_s \pi(a|s). \tag{36}$$

This vector represents the probability of choosing each of the three actions prior to encoding the current trial's stimulus. The degree of uncertainty over this prior on each trial is then computed via the Shannon entropy:

$$H_{\text{prior}} = -\sum \vec{\pi}^\mu \log_2 \vec{\pi}^\mu. \tag{37}$$

This quantity is then used to scale down the drift rates of each accumulator by the degree of uncertainty associated with taking any particular action in that trial:

$$v_i = \eta \left[ \frac{\pi(a_i|s)}{H_{\text{prior}}} \right]. \tag{38}$$

To generate RTs in the Standard RL and RLWM models, we draw each accumulator's starting point $k_i$ from a uniform distribution on the interval $[0, A]$. By randomly-sampling the starting point from some range of initial bias ($A$), we create sources of extra random variability that allow for the initial amount of evidence for each action $k_i$ to fluctuate from trial to trial. The drift rate of each accumulator $d_i$ is then drawn from a normal distribution, $N(v_i, s_v)$, where $v_i$ is calculated from Eq 38 using the respective $\pi$ for each model (i.e., Eq 28 for the Standard RL model and Eq 33 for the RLWM model). The drift rate $d_i$ represents the slope or speed of the accumulation process, which is proportional to the action probability specified by the current policy, according to Eq 38. The variance of the drift rate distribution, $s_v$, also serves to inject random trial-to-trial variability in the evidence accumulation process for each action.

On each trial, each accumulator's time to threshold $T_i$ can be computed via:

$$T_i = t_0 + \frac{B - k_i}{d_i}, \tag{39}$$

where $B$ is the LBA threshold bound and $t_0$ is the non-decision time that represents processes that are independent of the action choice, such as time for stimulus perception and response production. The agent's choice and corresponding RT on a single trial is determined by the accumulator that reaches threshold first (thereby generating the minimum RT):

$$a = \min(T) \tag{40}$$

**Modeling time pressure.** As mentioned in the Results, one of the main contributions of this study was modeling response time (RT) as a function of trial-by-trial policy cost and entropy (Eq 6). To account for the hypothesized effects of time pressure on choice behavior, we fit one separate parameter in the Adaptive models that was specific to Task 3. This parameter ($C_{\text{reduced}}$ for the Capacity and Capacity-Value models and $R_{\text{reduced}}$ for the Value model) modeled the effect of time pressure as a reduction in the agent's capacity limit ($C$) or aspiration level ($R$), forcing the agent to further compress their policies.

In the RLWM and Standard RL models, we assumed that time pressure would decrease the threshold bound of the accumulation process, and therefore we fit a separate bound parameter $B_2 - A$ (to ensure that $B_2 > A$) for the time pressure condition in Task 3. For the Adaptive models, we hypothesized that time pressure would further compress agents' policies by decreasing their capacity limit ($C$). We therefore fit a separate capacity parameter $C_{\text{reduced}}$ for the time pressure condition (Q2) in Task 3.

**Model variants.** To summarize, we consider the following policy compression model variants: the Fixed: $1\beta$ model (where we fit a single $\beta$ parameter across all tasks and conditions), the Adaptive: Capacity model, the Adaptive: Value model, and the Adaptive: Capacity-Value model. We also considered one additional variant of the Fixed model, where we fit a unique $\beta$ parameter for each condition in each task (3 tasks × 2 conditions per task = $6\beta$s). We called this the Fixed: $6\beta$ model. In total, we considered 5 variants of the policy compression model.

For the comparison models, we considered the RLWM model as well as two variants of the Standard RL model: one where we fit a single $\beta$ parameter across all tasks and conditions (the Standard RL: $1\beta$ model), and one where we fit a unique $\beta$ parameter for each condition in each task (the Standard RL: $6\beta$ model). In total, we considered 3 comparison models. All model variants that we considered, along with their free parameters, are summarized in Table 1.

**Table 1. List of models and their free parameters.** Models vary in whether they penalize policy complexity and how they update $\beta$.

| No. | Model | $\beta$ update rule | Parameters |
|-----|-------|---------------------|------------|
| 1 | RLWM | n/a | $C, \alpha_{RL}, \phi, \rho, \gamma, A, B_1, B_2, \eta$ |
| 2 | Standard RL: $1\beta$ | n/a | $\beta, \alpha, A, B_1, B_2, \eta$ |
| 3 | Standard RL: $6\beta$ | n/a | $\beta_{11}, \beta_{12}, \beta_{21}, \beta_{22}, \beta_{31}, \beta_{32}, \alpha, A, B_1, B_2, \eta$ |
| 4 | Fixed: $1\beta$ | n/a | $\beta, \alpha_\theta, \alpha_V, \alpha_P, b_1, b_2$ |
| 5 | Fixed: $6\beta$ | n/a | $\beta_{11}, \beta_{12}, \beta_{21}, \beta_{22}, \beta_{31}, \beta_{32}, \alpha_\theta, \alpha_V, \alpha_P, b_1, b_2$ |
| 6 | Adaptive: Capacity | $\Delta\beta = \alpha_\beta [C - \xi]$ | $C, C_{\text{reduced}}, \beta_0, \alpha_\beta, \alpha_\theta, \alpha_V, \alpha_P, b_1, b_2$ |
| 7 | Adaptive: Value | $\Delta\beta = \alpha_\beta [R - \rho]$ | $R, R_{\text{reduced}}, \beta_0, \alpha_\beta, \alpha_\theta, \alpha_V, \alpha_P, b_1, b_2$ |
| 8 | Adaptive: Capacity-Value | $\Delta\beta = \alpha_\beta \left[ \frac{C-\xi}{R-\rho} - \beta \right]$ | $C, C_{\text{reduced}}, R, \beta_0, \alpha_\beta, \alpha_\theta, \alpha_V, \alpha_P, b_1, b_2$ |

**Table 2. Parameter bounds for the RLWM and standard RL models.**

| Parameter | RLWM | Standard RL: $1\beta$ | Standard RL: $6\beta$ |
|---|---|---|---|
| $C$ | [2, 5] | - | - |
| $\beta$ | - | [1, 30] | all 6 $\beta$s $\in$ [1, 30] |
| $\alpha$ | [0, 1] | [0, 1] | [0, 1] |
| $\phi$ | [0, 1] | - | - |
| $\rho$ | [0, 1] | - | - |
| $\gamma$ | [0, 1] | - | - |
| $A$ | [0, 500] | [0, 500] | [0, 500] |
| $B_1 - A$ | [0, 500] | [0, 500] | [0, 500] |
| $B_2 - A$ | [0, 500] | [0, 500] | [0, 500] |
| $\eta$ | [0, 3] | [0, 3] | [0, 3] |
| Total # Parameters | 9 | 6 | 11 |

**Model fitting.** We used maximum likelihood estimation to jointly fit the choice and response time data for each subjects. The Standard RL and RLWM models were fit according to the methods described in [20]. Parameter constraints were defined according to Tables 2 and 3. In general, all learning rates were constrained in the range [0, 1] and the non-decision time $t_0$ was fixed to 150ms for all models.

We chose to fix several parameters. For the policy compression (Fixed and Adaptive) models, $\sigma$ was fixed to 0.9 for all models. For the RLWM and Standard RL models, the $s_v$ parameter was fixed at 0.1. Fixing this parameter has been shown to significantly improve LBA model identifiability [20, 42]. For the RLWM model, the inverse temperatures $\beta_{RL}$ and $\beta_{WM}$ were fixed at 50, consistent with previous studies [20, 23]. Since $\beta_{RL}$ and $\beta_{WM}$ do not have the same interpretation as the $\beta$ in the policy compression models, we chose to fix them to the values stated in the previous studies' best fitting model, as other parameters in the RLWM model were larger determinants of the model fit to data. Additionally, the RLWM model had a very large number of fitted parameters already (9), so adding even more would have contributed to potential degeneracy. We also note that changing the $\beta_{RL}$ and $\beta_{WM}$ parameters would not have changed the qualitative predictions of the RLWM model.

**Table 3. Parameter bounds for the policy compression models.**

| Parameter | Fixed: $1\beta$ | Fixed: $6\beta$ | Capacity | Value | Capacity-Value |
|---|---|---|---|---|---|
| $C$ | - | - | [lb, 3] | - | [lb, 3] |
| $C_{\text{reduced}}$ | - | - | [0, lb] | - | [0, lb] |
| $R$ | - | - | - | [0, 1] | [0, 1] |
| $R_{\text{reduced}}$ | - | - | - | [0, 1] | - |
| $\beta$ | [1, 30] | all 6 $\beta$s $\in$ [1, 30] | - | - | - |
| $\beta_0$ | - | - | [1, 10] | [1, 10] | [1, 10] |
| $\alpha_\theta$ | [0, 1] | [0, 1] | [0, 1] | [0, 1] | [0, 1] |
| $\alpha_V$ | [0, 1] | [0, 1] | [0, 1] | [0, 1] | [0, 1] |
| $\alpha_\beta$ | - | - | [0, 1] | [0, 1] | [0, 1] |
| $\alpha_p$ | [0, 1] | [0, 1] | [0, 1] | [0, 1] | [0, 1] |
| $b_1$ | [1, 500] | [1, 500] | [1, 500] | [1, 500] | [1, 500] |
| $b_2$ | [1, 500] | [1, 500] | [1, 500] | [1, 500] | [1, 500] |
| Total # Parameters | 6 | 11 | 9 | 9 | 10 |

All free parameters that we fit for each model are indexed in Tables 2 and 3.

**Parameter and model recovery.** We validated our modeling procedure in two ways. First, we assessed parameter recovery by refitting the data simulated from the winning "Adaptive: Capacity-Value" model and comparing the resulting parameter estimates to their ground truth. All 10 of the parameters exhibited reasonable parameter recoverability, with correlations ranging from 0.25 to 0.944 (mean r = 0.595; all statistically significant, $p < 0.0001$).

Second, we assessed model recovery by fitting the eight total model variants to the simulated data from the winning model and computing the Bayesian Information Criterion (BIC) and the protected exceedance probability (PXP) using Bayesian model comparison [43]. We first observed that the BICs of the three models that did not penalize policy complexity (RLWM, Standard RL ($1\beta$), and Standard RL ($6\beta$)) was significantly greater than the BICs of the policy compression models (mean $\Delta$ BIC = 7863.4), indicating a clear distinction between compression and non-compression models (S1 Fig). However, the policy compression model variants (Fixed ($1\beta$), Fixed ($6\beta$), Adaptive: Capacity, Adaptive: Value, and Adaptive: Capacity-Value) are less quantitatively distinguishable from one another, with the top 3 models within a BIC difference of only 71.7. Additionally, we found that the PXP could not accurately identify the data-generating model among the compression models. Regardless, we note that the BICs of the data-generating model were the most internally consistent (with a standard deviation of 281, compared to a mean SD = 561.56 for the other policy compression variants).

From this analysis, we can conclude that our modeling procedure is able to accurately distinguish between compression and non-compression models, but is less suited for identifying the correct model variant within the class of policy compression models. Designing experiments to distinguish between fixed and adaptive policy compression models is an avenue for further research.

## Ethics statement

Our study involved human subjects and was approved by the Harvard Institutional Review Board, number IRB15–2048. All subjects gave electronic written consent before beginning the study. We pre-registered our study and analyses at https://aspredicted.org/blind.php?x=ZZY_QBZ.

## Subjects

In accordance with a power analysis run on a pilot sample, we collected data from two-hundred (N = 200; 136 male) subjects who completed our study on Amazon Mechanical Turk and received monetary compensation. Subjects were paid a base pay of $4 and a performance bonus of up to $4 for completing the task. Subjects took, on average, 35 minutes to complete the entire experiment, and their average payout was $6.66. No subjects were excluded.

## Additional experiment details

All subjects completed 2 practice blocks of 30 trials each before beginning the actual experiment to familiarize them with the structure of the task. The practice blocks were identical to the experimental blocks in their trial-by-trial procedure, and the reward functions were designed to expose subjects to probabilistic reward and different stimulus-response contingencies. Each practice block had three unique stimuli and three unique key press responses (same as the experiment). Each stimulus was presented 10 times in one block. In one practice block, there was one optimal response for each stimulus, while in the other, all three stimuli shared one optimal action. They were allowed to return to this practice block as many times as they wanted throughout the study. We did not analyze data from these practice blocks.

In general, we tried to encourage independent learning of actions across states by informing the subjects that multiple states could share the same optimal action, or that one state could have more than one optimal action.

## Computing empirical policy complexity

Empirical policy complexity as plotted in the reward-complexity trade-off plots in Figs 4B, 5B and 7B was estimated from each subject's behavior per task condition. Following [12], we use the Hutter estimator, which computes the posterior expected value of the mutual information under a symmetric Dirichlet prior [44], to estimate subjects' empirical policy complexity, or the mutual information between the observed stimuli (states) and subjects' key press responses (actions).

## Supporting information

**S1 Fig. Model comparison.** The difference in Bayesian Information Criterion (BIC) relative to the model with the lowest BIC (Value). **(Inset)** A zoomed-in view of all the policy compression models. Policy compression models in general outperform the RLWM and Standard RL models. Error bars indicate standard error.
(PDF)

**S2 Fig. Dynamics of learning in Task 1 (manipulating state frequency). (A)** From left to right: The dynamic reward complexity trade-off, averaged across all subjects. Solid dot indicates the start, while open dot indicates the end, of learning. Policy complexity, average reward, and response time (RT) as a function of trials. Response time as a function of policy complexity. Note that policy complexity, average reward, and RT are computed via a sliding window of 30 trials. The running average in each plot is therefore truncated to 30 trials fewer than the total number of trials, as there are not enough elements to fill the window at endpoints. **(B)** Same as **(A)** but data simulated from the winning policy compression model (Adaptive: Capacity-Value). **(C)** Same as **(A)** but data simulated from the Adaptive: Value model. **(D)** Same as **(A)** but data simulated from the Adaptive: Capacity model. **(E)** Same as **(A)** but data simulated from the RLWM model. **(F)** Same as **(A)** but data simulated from the No Cost ($1\beta$) model. All shaded error bars indicate standard error.
(PDF)

**S3 Fig. Dynamics of learning in Task 2 (manipulating action frequency). (A)** From left to right: The dynamic reward complexity trade-off, averaged across all subjects. Solid dot indicates the start, while open dot indicates the end of learning. Policy complexity, average reward, and response time (RT) as a function of trials. Response time as a function of policy complexity. Note that policy complexity, average reward, and RT are computed via a sliding window of 30 trials. The running average in each plot is therefore truncated to 30 trials less than the total number of trials, as there are not enough elements to fill the window at endpoints. In the data, the policy complexity is similar across conditions and increases slightly over the course of the task. While the Capacity-Value model (B) captures this between-condition similarity, the Value (C) and Capacity (D) models predict diverging policy complexities. **(B)** Data simulated from the winning policy compression model (Adaptive: Capacity-Value). **(C)** Data simulated from the Adaptive: Value model. **(D)** Data simulated from the Adaptive: Capacity model. **(E)** Data simulated from the RLWM model. **(F)** Data simulated from the No Cost ($1\beta$) model. All shaded error bars indicate standard error.
(PDF)

**S4 Fig. Dynamics of learning in Task 3 (manipulating time pressure). (A)** From left to right: The dynamic reward complexity trade-off, averaged across all subjects. Solid dot indicates the start, while open dot indicates the end of learning. Policy complexity, average reward, and response time (RT) as a function of trials. Response time as a function of policy complexity. Note that policy complexity, average reward, and RT are computed via a sliding window of 30 trials. The running average in each plot is therefore truncated to 30 trials less than the total number of trials, as there are not enough elements to fill the window at endpoints. In the data, policy complexity remains roughly constant in both conditions, though it is overall higher in Q1 than in Q2. This trend is successfully mirrored by the Capacity-Value model (B). In the Value (C) model, the complexity difference between conditions is much smaller, while in the Capacity (D) model, complexity starts at the same point for both conditions and diverges with learning. Additionally, average reward steadily increases for both conditions, though it is always higher on average in Q1 than Q2. This overall difference in reward, although less pronounced, is captured by the Capacity-Value model but not by the Capacity and Value only models. **(B)** Data simulated from the winning policy compression model (Adaptive: Capacity-Value). **(C)** Data simulated from the Adaptive: Value model. (D) Data simulated from the Adaptive: Capacity model. **(E)** Data simulated from the RLWM model. **(F)** Data simulated from the No Cost ($1\beta$) model. All shaded error bars indicate standard error.
(PDF)

**S5 Fig. Example task conditions.** Two task conditions illustrating how the reward function affects the distribution of actions and changes the reward-complexity trade-off. **(A)** In this example condition, there is one unique rewarded action for each state. **(B)** This results in a roughly uniform marginal action distribution and a strictly monotonic reward-complexity trade-off. **(C)** In this condition, all states share the same rewarded action, causing the marginal action distribution to be heavily biased towards one action. **(D)** This reward structure results in a non-strictly monotonic reward-complexity trade-off. Note that in this condition, agents could achieve the highest average reward value with a variety of policy complexities. The example points on each plot show different suboptimal policies that can move in the reward-complexity space depending on how $\beta$ is being updated using the capacity limit $C$, aspiration level $R$, or a combination of both.
(PDF)

## Acknowledgments

We are thankful to Roey Schurr and other members of the Computational Cognitive Neuroscience Laboratory for helpful comments. This work was partially conducted while visiting the Okinawa Institute of Science and Technology (OIST) through the Theoretical Sciences Visiting Program (TSVP).

## Author Contributions

**Conceptualization:** Lucy Lai, Samuel J. Gershman.

**Data curation:** Lucy Lai.

**Formal analysis:** Lucy Lai, Samuel J. Gershman.

**Funding acquisition:** Lucy Lai, Samuel J. Gershman.

**Investigation:** Lucy Lai, Samuel J. Gershman.

**Methodology:** Lucy Lai, Samuel J. Gershman.

**Project administration:** Lucy Lai, Samuel J. Gershman.

**Resources:** Lucy Lai, Samuel J. Gershman.

**Software:** Lucy Lai.

**Supervision:** Lucy Lai, Samuel J. Gershman.

**Validation:** Lucy Lai.

**Visualization:** Lucy Lai.

**Writing – original draft:** Lucy Lai, Samuel J. Gershman.

**Writing – review & editing:** Lucy Lai, Samuel J. Gershman.

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
