## [Decision Letter · Decision Letter 0]

11 Jan 2024

Dear Ms. Lai,

Thank you very much for submitting your manuscript "Human decision making balances reward maximization and policy compression" for consideration at PLOS Computational Biology.

As with all papers reviewed by the journal, your manuscript was reviewed by members of the editorial board and by several independent reviewers. In light of the reviews (below this email), we would like to invite the resubmission of a significantly-revised version that takes into account the reviewers' comments.

As you will see from the comments below the revewers did have some suggestions to make, some of these minor, some major. Most of the suggestions require rewriting parts of the article, although it is also possible some more computational modeling may be required to address the reviewer comments.

We cannot make any decision about publication until we have seen the revised manuscript and your response to the reviewers' comments. Your revised manuscript is also likely to be sent to reviewers for further evaluation.

Sincerely,

Ulrik R. Beierholm

Academic Editor

PLOS Computational Biology

Thomas Serre

Section Editor

PLOS Computational Biology

Reviewer's Responses to Questions

**Comments to the Authors:**

Reviewer #1: In their manuscript, 'Human Decision Making Balances Reward Maximization and Policy Compression,' the authors explore decision-making and learning processes, and their underlying cognition in humans, by developing a new policy-compression theory. According to the theory, in a reinforcement-learning (RL) context, the mapping between states and actions, the policy, is characterized by complexity, the extent to which action selection is state-specific vs. shared across states. Considering the limited capacity of human decision-makers' policy complexity, the authors use information-theory formalism to test several hypotheses regarding the expected reward, stochasticity, and reaction time that would be realized under different conditions. In three novel experimental tasks with a within-subject design, the authors modify the distribution of stimuli (states) presentation frequency to be more or less uniform (task 1), the extent to which optimal actions are shared across states (task 2), and the time allotted for decision making (task 3). Across these tasks, and using previously published data, the authors find support to key predictions of the policy-compression theory. The policy compression theory fares well in quantitative model-comparison measures, as well as replicates qualitative behavioral patterns. Taken together, this evidence suggests that the policy compression model goes above and beyond the predictions of standard reinforcement learning (RL) models, as well as RL models limited by a working memory capacity (RLWM).

I really enjoyed reading this well-written manuscript. The elegant development of the policy-complexity theory and its application to human cognition makes an important contribution, offering a consolidated framework for theories from different domains, including bounded rationality and reinforcement learning. By doing so, the authors substantially promote a novel understandings of human cognition.

My one main comment is on the focus and the organization of the manuscript. While I believe that addressing this comment may assist future readers, this is largely a matter of style. So, I only offer this as a suggestion and leave it to the authors' discretion.

In its current form, the manuscript gives similar weight to the presentation of the policy compression theory, which was partially developed in previous publications, and the novel experiments with their respective results. I highly appreciated the comprehensive presentation of the theory, allowing this manuscript consistent notation and having it as a self-contained unit. However, for me, a greater focus on the experiments and their results would have made the reading clearer while motivating the reader to seek a deeper understanding of the theory. Specifically, the Introduction does a great job of presenting relevant background but uses a lot of terminology that the naive reader may not be familiar with. Such presentation requires the readers to invest in familiarizing themselves with substantial new terminology, without yet knowing which parts of this theory would be relevant and why. Similarly, sections 2.1-2.4 present, very clearly, the policy-compression theory, but the reader does not know yet to what end. An alternative approach could be to use the Introduction for a more intuitive presentation of the basic concepts of policy compression, and how these may be realized under the different conditions tested in tasks 1-3. Such focus on hypotheses may allow the general reader a fluent understanding of the main concepts. Restructuring the Results section to present the experiments and their results upfront would allow general readers to quickly grasp the practical implications and utility of the theory, without the immediate need to understand the formal aspect of the theory, while still offering interested readers the in-depth theoretical exposition in subsequent sections.

Minor comments:

1. Complexity as a feature of participants (e.g., "individuals with low complexity") is mentioned throughout the text but I was not sure exactly what does that mean – is it the per-block policy complexity inferred from behavior?

2. In task 3 the authors test the effects of time pressure on policy compression. I wondered whether the model could provide some quantitative insights into the cognitive mechanisms involved in the hypothesized compression. Namely, whether it is possible within the framework of the specific model used to estimate the improvement in compression per unit of time.

3. In the following sentence and throughout the text, the manuscript interprets reduced complexity under time pressure as a feature of time-dependent decoding: “… individuals reduce their policy complexity under time pressure, providing evidence for the hypothesis that actions are selected through time-sensitive decoding of a compressed code”. However, if I understood correctly, in the presented tasks decision-making was intertwined with learning. So, a priori, time pressure may affect the learning part as well (policy formation and its encoding). I wondered if there was a reason to focus the interpretation on the decoding part.

4. The manuscript outlines a spectrum of policies varying from policies with state-specific actions (higher complexity) to policies with actions that shared across states (lower complexity), which appears to align with the distinction between model-based and model-free reinforcement learning. A discussion on the intersection between these theories and potential implications may provide a broader context for interpreting the results.

5. Minor errors in the text:

a. Figure 2: the caption might be confusing left and right.

b. The following seem grammatically incorrect:

i. "This parameter … assumes that time pressure would compress agents’ policies via a decrease their capacity limit C or aspiration level R for that condition."

ii. "we hypothesized that time pressure would compress agents’ policies via a decrease their capacity limit C for that condition"

----

Signed: Ohad Dan

Reviewer #2: review of Lai & Gershman: “Human decision making balances reward maximization and policy compression”

This study describes a set of experiments testing a computational model of learning under cognitive computational constraints, called “policy compression”. This model views learning as a rate-distortion problem of finding the maximal obtainable Value of a learned policy under a constraint on its Complexity. As in standard reinforcement learning models, the policy is defined as a stochastic mapping from states to actions and its Value is the expected accumulated reward. Building on an information-theoretic learning framework previously developed by Tishby and others, the authors define a policy's Complexity as the mutual-information between states and actions, i.e., how much information, on average, does an agent’s selected action contain about the underlying state eliciting it.

The main findings are that human choice behavior and reaction times in instrumental learning tasks are better explained by the policy-compression model compared to other reinforcement learning based models. Specifically, people are able to utilize statistical redundancy in their environment (state, action and reward distributions) to compress their policies, i.e., reduce their complexity) resulting in preservative behavior that is biased towards frequently selected actions. They also found that policy complexity is reduced under time pressure, based on the hypothesis that action-selection requires a time-dependent decoding of the compressed state representation.

Overall, this is an elegant contribution to the growing literature demonstrating the usefulness of information-theoretic principles for elucidating the mechanisms and trade-offs underlying learning and cognition. My main concerns and more substantial comments are detailed under “General comments” below, whereas smaller issues and corrections appear “Minor points”.

General comments:

The issue of learning dynamics, namely how does the value-complexity tradeoff evolve throughout learning, deserves a more detailed treatment in the main-text, rather than being relegated to the supplementary (Figs. S2-S4). For example, in several occasions the authors mention “individuals with low complexity” (lines 78, 557) but it is likely that individual’s complexity evolves throughout the learning process and it would be interesting to see, for example, whether different subjects exhibit different learning dynamics, i.e., different beta update profiles, throughout learning. In particular, previous studies have reported a non-monotonic evolution profile of the policy complexity, in both biological and artificial learning settings, with an emphasis on value optimization at the early stage of learning shifting to compression at a later stage. A similar trend may be seen, at least in tasks 1 and 2, for the Policy Compression (Adaptive: Value) model (Figs. S2-S3) but not for the other models or data (or in task 3). It would be interesting if the authors could comment on this effect and why it appears only in this model variant.

Since the marginal action probability, P(a), depends on the policy (and vice-versa), it would be helpful to clarify with respect to which policies are the marginal distributions in Figs. 4-6 (subfigures (A)) computed. Are they derived from the optimal unbounded policy (with beta=infinity) or from a policy with a finite beta value estimated from the data? More importantly, this circular dependency between the policy and the marginal action distribution can lead to degeneracies in the definition of the optimal tradeoff. For example, there are multiple state-independent action distributions with zero complexity (delta function, uniform etc.). Does the theory make any specific predictions regarding such cases in which the “optimal” policy may be ill-defined?

Why do the authors compare their model with the RLWM model rather than other models which may seem more immediately related to the policy compression framework such as entropy-regularized RL or “empowerment” related frameworks? Alternatively, shouldn't beta_RL and beta_WM should be allowed to change (or at least be fitted) in analogy with the role of beta in the policy compression model? The authors should clarify the motivation for this choice and/or provide comparison with other, more immediately related models.

The authors write (lines 96-98) that the theoretical framework of policy optimization under information-theoretic constraints was originally developed in their previous work (Lai, Gershman 2021, Gershman, Lai 2021). However, this attribution might be (unintentionally) misleading as such frameworks were developed at least a decade earlier, e.g. in (Tishby & Polani, 2010), (Parush, Tishby & Bergman 2011), that are indeed cited by the authors but in a different context. Also (Rubin, Shamir & Tishby 2012) develops a capacity constrained policy optimization framework in a more general setting of Markov Decision Processes. The authors should reference these earlier frameworks and elucidate in what ways they build upon or extend them (e.g. via the actor-critic process model or link to RT).

Eq. 8 and lines 129-13: shouldn’t this relation hold for all points on the optimal reward-complexity trade-off curve (rather than just at the capacity constraint I(S;A)=C)?

Eq. 10: the authors should clarify what assumptions (if any) are implied by choosing this specific parametric form for the class of admissible policies. Is it guaranteed to converge to the optimal solution of the variational problem (Eq. 6)? Also, theta_(s,a) seems to be a proxy for Q(s,a). If so, why is it updated by the full policy cost (Eq. 11), including the complexity term rather than just the reward? Similarly, why is theta updated using an actor-critic model whereas P(a), xi and rho are updated via an exponential moving average. Finally, why are only the alpha_xi and alpha_rho learning parameters specified as fixed (at 0.01)?

Eq. 19: shouldn’t beta on the r.h.s be replaced with 1? otherwise it is not clear to me how this results in the update dynamics described in the text below.

Fig. 4B: the policy complexity difference between the Q1 and Q2 conditions in the Policy compression model seem quite small, which perhaps explains why no significant difference in complexity was found in the data. Why didn’t the experimental design include higher policy complexity contrast conditions?

Fig. 5B: how can the policy complexity level differ between Q1 and Q2 in the Standard RL & RLWM models (which are not supposed to be complexity sensitive)?

Figure 8B: the blue and red lines seem almost overlapping but shouldn’t there be overall lower complexity for the Q2 condition in task 2?

Minor points:

Eq. 13: V is missing a caret.

Fig. 2 caption text: (Right) and (Left) should be switched (also, “monotonic” and “non-monotonic” should be "

---

## [Decision Letter · Decision Letter 1]

8 Apr 2024

Dear Ms. Lai,

We are pleased to inform you that your manuscript 'Human decision making balances reward maximization and policy compression' has been provisionally accepted for publication in PLOS Computational Biology.

Best regards,

Thomas Serre

Section Editor

PLOS Computational Biology

Thomas Serre

Section Editor

PLOS Computational Biology

Reviewer's Responses to Questions

**Comments to the Authors:**

Reviewer #1: I thank the authors for their thorough and thoughtful point-by-point responses to my comments. It is evident that considerable effort was employed in addressing the feedback from the reviewers, and this has significantly enhanced the clarity of your work. I appreciate the overall reorganization of the manuscript and believe it substantially contributes to its coherence.

I found the model overview presented in section 2.1 to be notably concise and informative, effectively setting the stage for the ensuing presentation of the experimental results. Additionally, the revised Figures 1 and 2 are well-designed and elucidate the basic theoretical framework and the key predictions of the study, making the main concepts more accessible to the reader. Finally, I valued the addition of the individual differences perspective in section 2.3.

I congratulate the authors on this successful revision and have no further comments on the manuscript.

Reviewer #2: The authors have adequately addressed the main concerns raised in the review.

Congratulations on an interesting and well executed work!

**Have the authors made all data and (if applicable) computational code underlying the findings in their manuscript fully available?**

Reviewer #1: Yes

Reviewer #2: Yes

PLOS authors have the option to publish the peer review history of their article (what does this mean?). If published, this will include your full peer review and any attached files.

Reviewer #1: **Yes: **Ohad Dan

Reviewer #2: **Yes: **Nadav Amir

---

## [Editor Report · Acceptance letter]

19 Apr 2024

PCOMPBIOL-D-23-01913R1 

Human decision making balances reward maximization and policy compression

Dear Dr Lai,

I am pleased to inform you that your manuscript has been formally accepted for publication in PLOS Computational Biology. Your manuscript is now with our production department and you will be notified of the publication date in due course.

With kind regards,

Anita Estes
